synthetic chemistry

theophylline, corrosion inhibition, API 5 L X52 steel, acid, quantum chemical calculation

**Author for correspondence:**
Guillermo E. Negrón-Silva
e-mail: gns@azc.uam.mx

Subsequent to acceptance, the authors indicated a number of minor changes to their manuscript were required. These changes were assessed and approved by the Editors prior to publication. The authors have made the preprint (i.e. initial submission version) and postprint (i.e. accepted but not typeset version) available at https://eartharxiv.org/jvkqr/ and https://eartharxiv.org/7nz9h, respectively, in the interests of transparency.

This article has been edited by the Royal Society of Chemistry, including the commissioning, peer review process and editorial aspects up to the point of acceptance.

# Adsorption and corrosion inhibition behaviour of new theophylline–triazole-based derivatives for steel in acidic medium

Araceli Espinoza-Vázquez[1], Francisco Javier Rodríguez-Gómez[1], Ivonne Karina Martínez-Cruz[2], Deyanira Ángeles-Beltrán[2], Guillermo E. Negrón-Silva[2], Manuel Palomar-Pardavé[3], Leticia Lomas Romero[3,4], Diego Pérez-Martínez[3,4] and Alejandra M. Navarrete-López[2]

[1]Facultad de Química, Departamento de Ingeniería Metalúrgica, Universidad Nacional Autónoma de México, Av. Universidad No. 3000, Coyoacán, C.U., Ciudad de México, C.P. 04510, Mexico
[2]Departamento de Ciencias Básicas, and [3]Departamento de Materiales, Universidad Autónoma Metropolitana-Azcapotzalco, Av. San Pablo No. 180, Ciudad de México, C.P. 02200, Mexico
[4]Departamento de Química, Universidad Autónoma Metropolitana-Iztapalapa, Av. San Rafael Atlixco No. 186, Ciudad de México, C.P. 09340, Mexico

GEN-S, 0000-0002-7886-5261; DP-M, 0000-0003-2907-8138

The design and synthesis of a series of theophylline derivatives containing 1,2,3-triazole moieties are presented. The corrosion inhibition activities of these new triazole–theophylline compounds were evaluated by studying the corrosion of API 5 L X52 steel in 1 M HCl medium. The results showed that an increase in the concentration of the theophylline–triazole derivatives also increases the charge transference resistance ($R_{ct}$) value, enhancing inhibition efficiency and decreasing the corrosion process. The electrochemical impedance spectroscopy under static conditions studies revealed that the best inhibition efficiencies (approx. 90%) at 50 ppm are presented by the all-substituted compounds. According to the Langmuir isotherm, the compounds 4 and 5 analysed exhibit physisorption–chemisorption process, with exception of the hydrogen 3, bromo 6 and iodo 7 substituted compounds, which exhibit chemisorption process. The corrosion when submerging a steel bar in 1 M HCl was studied using SEM-EDS. This experiment showed that the corrosion process decreases considerably in the presence of 50 ppm of the organic inhibitors.

Finally, the theoretical study showed a correlation between $E_{HOMO}$, hardness ($\eta$), electrophilicity (W), atomic charge and the inhibition efficiency in which the iodo **7** substituted compound presents the best inhibitor behaviour.

# 1. Introduction

Steel corrosion remains as one of the most significant problems to industry. This naturally occurring phenomenon, that takes place at the metal–solution interface, substantially decreases the life of the equipment and facilitates the dissolution of environmentally toxic metal from the components [1–3]. In this regard, several organic molecules are recognized as corrosion inhibitors for a number of metals and alloys [4–6]. The adsorption of organic molecules at the metal surface both disrupts the properties of the metal/solution interface, effectively inhibiting the corrosion process [7], and eliminates the need for expensive and toxic inhibitor compounds [8].

It has been found that molecules with lone electron pairs and/or π-electrons in their structure show great affinity to be adsorbed into the metallic material [9–12]. Thus, heterocycles containing nitrogen, oxygen, sulfur and unsaturation in their structure are excellent candidates to be evaluated as corrosion inhibitors. In this context, nitrogen-containing heterocycles have been regarded as the most effective corrosion inhibitors of steel in acid solutions [13,14].

Additionally, to their biological activities [15–21] nitrogen-containing heterocycles derived from xanthine, such as theophylline and theobromine, have shown corrosion inhibition activity (although these promote corrosion under certain conditions) [22,23]. The caffeine has demonstrated to possess great activity as corrosion inhibitor for a series of metals and alloys.

Recently, theophylline has been studied as a corrosion inhibitor using an API 5 L X70 steel and proved effective at low concentrations [24].

On the other hand, the triazole derivatives are another class of nitrogen-containing heterocycles that have drawn attention for their potential applications in pharmaceuticals, coordination chemistry and as corrosion inhibitors. Several reports have highlighted the capabilities of these compounds to strongly adsorb on metal surfaces, achieving an adequate corrosion inhibition efficiency at low concentrations [25–32].

A typical pathway to enhance the corrosion inhibition efficiency of a given heterocycle is to modify its structure with various substituents or functional groups. Moreover, a synergistic effect could be achieved if those substituents possess corrosion inhibition activity by themselves. Accordingly, this work presents the design and synthesis of new theophylline derivatives bearing 1,2,3-triazole moieties, those corrosion inhibitors that are easy to prepare and low cost. These novel compounds were tested as corrosion inhibition species in order to establish structure–activity relationships and to gain further insight into their adsorption properties and the steel protection.

# 2. Material and methods

## 2.1. Synthesis of the theophylline–triazole inhibitors (figure 1)

### 2.1.1. 1,3-dimethyl-7-(prop-2-in-1-yl)-3,7-dihydro-1H-purine-2,6-dione (2)

Compound **2** was synthesized following the procedure described by Ruddarraju *et al.* [21]. A mixture of theophylline (**1**) (1.98 g, 11 mmol) and potassium carbonate (1.990 g, 14.4 mmol) in DMF (30 ml) were stirred vigorously at room temperature for 20 min. After this time, propargyl bromide (1.68 ml, 22.2 mmol) was added and temperature was increased at 85°C with vigorous stirring for another 2 h. Then, the mixture was poured in cold water. The compound was recovered as a white powder: yield 80%, m.p. 220–222°C. $^{1}$H NMR (500.13 MHz, CDCl$_3$): $\delta$ (ppm) = 2.60 (1H, t, J = 2.61 Hz, H12), 3.41 (3H, s, N1-CH$_3$), 3.60 (3H, s, N3-CH$_3$), 5.17 (2H, dd, J = 2.6, 0.62 Hz, H10), 7.83 (1H, t, J = 0.53 Hz, H8). $^{13}$C NMR (125.77 MHz, CDCl$_3$): $\delta$ = 27.96 (N1-CH$_3$), 29.79 (N3-CH$_3$), 36.44 (C10), 75.43 (C11), 76.07 (C12), 106.71 (C5), 140.42 (C8), 148.92 (C4), 151.60 (C2), 155.23 (C6). FT-IR/ATR vmax/cm$^{-1}$: 3243.55, 3111.71, 2946.11, 2127.13, 1703.88, 1651.15, 1543.95, 1477.34, 1437.21, 1373.59, 1232.32, 1190.89, 1025.01, 977.45, 744.20.

### 2.1.2. 7-((1-benzyl-1H-1,2,3-triazol-4-yl) methyl)-1,3-dimethyl-3,7-dihydro-1H-purine-2,6–dione (3)

A mixture of compound **2** (206 mg, 1 mmol), sodium ascorbate (40 mg, 0.2 mmol), sodium azide (78 mg, 1.2 mmol), benzyl chloride (0.14 ml, 1.2 mmol) and Cu/Al-mixed oxide (40 mg) in 6 ml of ethanol/

**Figure 1.** Synthesis of 1,2,3-triazoles in the presence of Cu(Al)O.

| compound | R | X |
|----------|-----|------|
| **3** | –H | –Cl |
| **4** | –F | –Cl |
| **5** | –Cl | –Cl |
| **6** | –Br | –Br |
| **7** | –I | –Br |

water (3:1) were stirred at 80°C for 30 min with microwave radiation. After this time, the Cu/Al-mixed oxide is recovered by centrifugation and the supernatant is poured in 20 ml of water, extracted with dichloromethane and dried over sodium sulfate anhydrous. Compound **3** is obtained, after chromatographic purification (CH$_2$Cl$_2$:EtOH 95:5), as a white powder: yield 78%, m.p. 169–171°C. [1]H NMR (500.13 MHz, CDCl$_3$): δ = 3.38 (3H, s, N1-CH$_3$), 3.56 (3H, s, N3-CH$_3$), 5.49 (2H, s, H13), 5.56 (2H, s, H10), 7.26 (2H, m, H15), 7.36 (3H, m, H17, H16), 7.75 (1H, s, H12), 7.81 (1H, s, H8). [13]C NMR (125.77 MHz, CDCl$_3$): δ = 27.98 (N1-CH$_3$), 29.81 (N3-CH$_3$), 41.48 (C10), 54.32 (C13), 106.45 (C5), 123.48 (C12), 128.09 (C15), 128.89 (C17), 129.15 (C16), 134.23 (C14), 141.32 (C8), 142.52 (C11), 148.93 (C4), 151.58 (C2), 155.40 (C6). FT-IR/ATR vmax/cm$^{-1}$: 3114.70, 2957.28, 1690.39, 1650.26, 1546.81, 1453.58, 1214.63, 1021.75, 749.91. HRMS (ESI-TOF) (calculated for C$_{17}$H$_{18}$N$_7$O$_2$ + H+): 352.1516; found: 352.1514.

### 2.1.3. 7-((1-(4-fluorobenzyl)-1H-1,2,3-triazol-4-yl) methyl)-1,3-dimethyl-3,7-dihydro-1H-purine-2,6-dione (**4**)

Compound **4** was synthesized following the procedure described previously for compound **3**, from compound **2** and 4-fluorobenzyl chloride. Compound **4** is obtained, after chromatographic purification (CH$_2$Cl$_2$:EtOH 95:5), as a white powder: yield 90%, m.p. 184–186°C. [1]H NMR (400.13 MHz, CDCl$_3$): δ = 3.39 (3H, s, N1-CH$_3$), 3.56 (3H, s, N3-CH$_3$), 5.47 (2H, s, H13), 5.56 (2H, s, H10), 7.06 (2H, t, J = 8.61 Hz, H15), 7.26 (2H, dd, J = 8.64, 4.34 Hz, H16), 7.75 (1H, s, H12), 7.82 (1H, s, H8). [13]C NMR (100.61 MHz, CDCl$_3$): δ = 27.99 (N1-CH$_3$), 29.82 (N3-CH$_3$), 41.47 (C10), 53.58 (C13), 106.45 (C5), 116.09 (C15), 116.31 (C15), 123.39 (C12), 129.98 (C16), 130.06 (C16), 141.35 (C8), 142.65 (C11), 148.99 (C4), 151.59 (C2), 155.44 (C6), 161.69 (C14 or C17), 164.17 (C14 or C17). FT-IR/ATR vmax/cm$^{-1}$: 3144.88, 3116.27, 3000.48, 2960.31, 1691.08, 1651.91, 1549.09, 1512.06, 1456.51, 1226.98, 1023.54, 786.67, 750.59, 615.31, 522.38. HRMS (ESI-TOF) (calculated for C$_{17}$H$_{17}$N$_7$O$_2$F + H+): 370.1422; found: 370.1419.

### 2.1.4. 7-((1-(4-chlorobenzyl)-1H-1,2,3-triazol-4-yl) methyl)-1,3-dimethyl-3,7-dihydro-1H-purine-2,6-dione (**5**)

Compound **5** was synthesized following the procedure described for compound **3**, from compound **2** and 4-chlorobenzyl chloride. Compound **5** is obtained, after chromatographic purification (CH$_2$Cl$_2$: EtOH 95:5), as a light green powder: yield 76%, m.p. 194–196°C. [1]H RMN (400.13 MHz, CDCl$_3$): δ = 3.39 (3H, s, N1-CH$_3$), 3.56 (3H, s, N3-CH$_3$), 5.47 (2H, s, H13), 5.56 (2H, s, H10), 7.20 (2H, d, J = 8.42 Hz, H15), 7.34 (2H, d, J = 842. Hz, H16), 7.77 (1H, s, H12), 7.82 (1H, s, H8). [13]C NMR (100.61 MHz, CDCl$_3$): δ = 27.98 (N1-CH$_3$), 29.81 (N3-CH$_3$), 41.45 (C10), 53.58 (C13), 106.45 (C5), 123.53 (C12), 129.38 (C15), 129.44 (C16), 132.72 (C14), 135.01 (C17), 141.36 (C8), 142.77 (C11), 149 (C4), 151.58 (C2), 155.44 (C6). FT-IR/ATR vmax/cm$^{-1}$: 3096.88, 3052.15, 2960.28, 1688.25, 1650.83, 1555.24, 1406.79, 1220.94, 1082.17, 1045.89,

978.31, 848.97, 785.63, 770.92, 608.01, 494.81. HRMS (ESI-TOF) (calculated for $C_{17}H_{17}N_7O_2Cl + H+$): 386.1127; found: 386.1124.

### 2.1.5. 7-((1-(4-bromobenzyl)-1H-1,2,3-triazol-4-yl) methyl)-1,3-dimethyl-3,7-dihydro-1H-purine-2,6-dione (6)

Compound **6** was synthesized following the procedure described for compound **3**, from compound **2** and 4-bromobenzyl bromide. Compound **6** is obtained, after chromatographic purification ($CH_2Cl_2$:EtOH 95:5), as a white powder: yield 63%, m.p. 199–201°C. $^1H$ RMN (500.13 MHz, $CDCl_3$): $\delta = 3.39$ (3H, s, N1-$CH_3$), 3.56 (3H, s, N3-$CH_3$), 5.45 (2H, s, H13), 5.56 (2H, s, H10), 7.13 (2H, d, J = 8.65 Hz, H15), 7.49 (2H, d, J = 8.61 Hz, H16), 7.76 (1H, s, H12), 7.80 (1H, s, H8). $^{13}C$ NMR (125.77 MHz, $CDCl_3$): $\delta = 27.96$ (N1-$CH_3$), 29.79 (N3-$CH_3$), 41.46 (C10), 53.62 (C13), 106.45 (C5), 123.09 (C17), 123.49 (C12), 129.69 (C15), 132.33 (C16), 133.24 (C14), 141.31 (C8), 142.81 (C11), 149 (C4), 151.56 (C2), 155.42 (C6). FT-IR/ATR $vmax/cm^{-1}$: 3134.60, 3095.80, 3049.46, 2960.09, 1687.19, 1650.07, 1554.67, 1450.68, 1405.55, 1220.14, 1103.66, 1030.53, 978.62, 847.88, 607.67, 488.92. HRMS (ESI-TOF) (calculated for $C_{17}H_{17}N_7O_2Br + H+$): 430.0621; found: 430.0618.

### 2.1.6. 7-((1-(4-iodobenzyl)-1H-1,2,3-triazol-4-yl) methyl)-1,3-dimethyl-3,7-dihydro-1H-purine-2,6-dione (7)

Compound **7** was synthesized following the procedure described for compound **3**, from compound **2** and 4-iodobenzyl bromide. Compound **7** is obtained, after chromatographic purification ($CH_2Cl_2$:EtOH 95:5), as a white powder: yield 68%, m.p. 181–184°C. $^1H$ NMR (400.13 MHz, $CDCl_3$): $\delta = 3.39$ (3H, s, N1-$CH_3$), 3.56 (3H, s, N3-$CH_3$), 5.44 (2H, s, H13), 5.56 (2H, s, H10), 7.00 (2H, d, J = 8.38 Hz, H15), 7.69 (2H, d, J = 8.37 Hz, H16), 7.76 (1H, s, H12), 7.82 (1H, s, H8). $^{13}C$ NMR (100.61 MHz, $CDCl_3$): $\delta = 28.02$ (N1-$CH_3$), 29.84 (N3-$CH_3$), 41.46 (C10), 53.73 (C13), 94.76 (C17), 106.44 (C5), 123.54 (C12), 129.88 (C15), 133.88 (C14), 138.29 (C16), 141.35 (C8), 142.72 (C11), 148.98 (C4), 151.58 (C2), 155.43 (C6). FT-IR/ATR $vmax/cm^{-1}$: 3143.55, 3115.45, 2958.37, 1690.45, 1651.19, 1548.13, 1456.04, 1218.88, 1006.66, 750.30, 615.40, 498.95. HRMS (ESI-TOF) (calculated for $C_{17}H_{17}N_7O_2I + H+$): 478.0483; found: 478.0478.

## 2.2. Corrosion inhibition tests

### 2.2.1. API 5 L X52 steel

API 5 L X52 steel was used for the corrosion inhibition studies. This type of steel has a metallographic preparation with the following nominal composition (wt%): C, 0.025; Mn, 1.65; Si, 0.26; Ti, 0.015; V, 0.001; Nb, 0.068; Mo, 0.175; S, 0.0025; Al, 0.045; Ni, 0.08; Cr, 0.07; Cu, 0.21; and balance iron.

### 2.2.2. Inhibitor solutions

A 0.01 M solution of each theophylline–triazole inhibitor **3–7** in DMF was prepared. Then, concentrations of 5, 10, 20 and 50 ppm of the inhibitor were added to the 1 M HCl corrosive solution.

### 2.2.3. Characterization of surfaces by SEM-EDS

The API 5 L X52 steel surface was prepared both without (blank) and with inhibitor; a 50 ppm concentration was used for a 24 h immersion time. After that experiment, the steel was washed with distilled water, dried and the surface analysed using a Zeiss SUPRA 55 VP electronic sweep microscope at 10 kV with a 300× secondary electron detector.

### 2.2.4. Electrochemical evaluation

The potential was stabilized at 20°C for approximately 1800 s before electrochemical impedance spectroscopy (EIS) test. EIS: a sinusoidal potential of $\pm 10$ mV was applied in a frequency interval of $10^{-2}$–$10^4$ Hz, in an electrochemical cell with three electrodes using Gill Ac. The working electrode was API 5 L X52 steel, while reference electrode and counter electrode were Ag/AgCl saturated with chloride potassium and graphite, respectively. The electrode surface was prepared using conventional metallography methods over an exposed area of 1 $cm^2$. After EIS measurements, the potentiodynamic polarization curves of 5 and 50 ppm of inhibitors were obtained. The measurements covered a range from $-500$ to 500 mV regarding the open circuit potential (OCP), with a sweep velocity of 60 mV $min^{-1}$ using the ACM Analysis software for data interpretation.

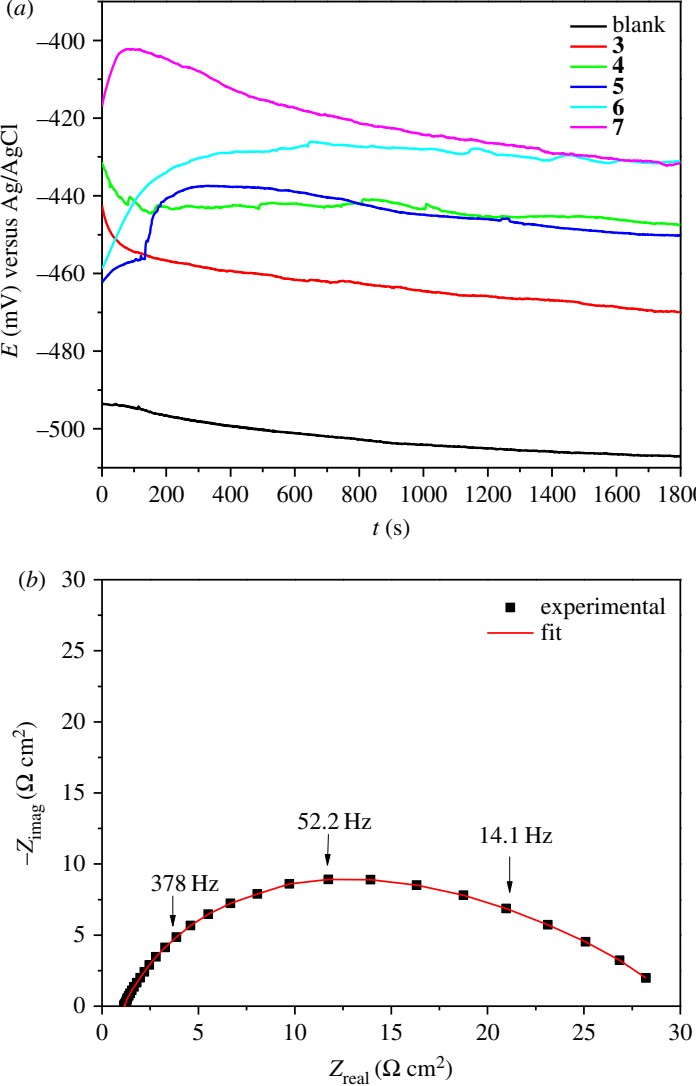

**Figure 2.** (a) Open circuit potential (OCP) versus time of theophylline–triazoles derivatives at 50 ppm and (b) Nyquist diagram without inhibitor in immerse API 5 L X52 steel.

### 2.2.5. Theoretical assessment

The calculations have been performed with Gaussian09 [33] using the M06–2X functional [34] and LANL2DZ basis set. For Cl, Br and I the LANL2 effective core potential was coupled with the LANL2DZ basis set. Frequency calculations were executed in order to guarantee the minimal energy structure. The solvent (water) effect is including with the solvation model based on density (SMD) [35].

## 3. Results and discussion

### 3.1. OCP and EIS electrochemical evaluation

Figure 2a shows the variations in time of the open circuit potential (OCP) in static conditions at 50 ppm, at the electrode made of API 5 L X52 steel in the presence of different inhibitor concentrations. It should be mentioned that it was necessary to stabilize the OCP before doing the EIS determinations. The steady state was reached after 1700 s. Figure 2b corresponds to the Nyquist diagram for the system without inhibitor, which depicts a depressed semicircle reaching a maximum $Z_{real}$ value of 30 Ω cm$^2$ (adjusted with electrical circuit figure 3a).

The Nyquist diagrams of each of the theophylline–triazoles are shown in figure 4a–e. As can be seen, the diameter of the semicircle increases proportionally with the inhibitor's concentration.

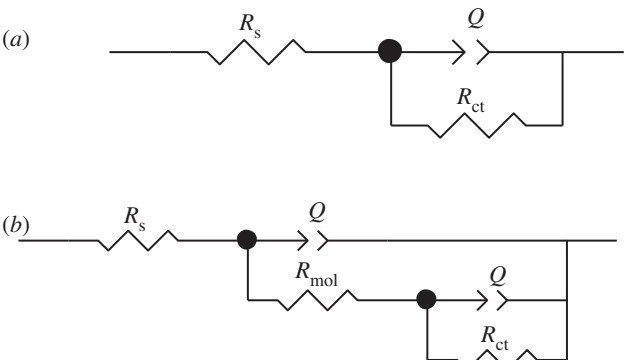

**Figure 3.** Equivalent electric circuit used in the system with (*b*) and without inhibitor (*a*).

**Figure 4.** Nyquist diagrams of theophylline–triazoles in immerse API 5 L X52 steel at different inhibitor concentrations. (*a–e*) Corresponds to compounds **3**, **4**, **5**, **6** and **7** respectively.

**Table 1.** Electrochemical parameters of 1,2,3-triazoles 1,4-disubstituted in API 5 L X52 steel immersed in HCl 1 M.

| inhibitor | C (ppm) | $R_s$ ($\Omega$ cm$^2$) | n | $C_{dl}$ ($\mu$F cm$^{-2}$) | $R_{ct}$ ($\Omega$ cm$^2$) | $R_{mol}$ ($\Omega$ cm$^2$) | $\eta$ (%) | $\chi^2$ |
|---|---|---|---|---|---|---|---|---|
| blank | 0 | 0.8 | 0.8 | 310.0 | 30 | — | — | — |
| **3** | 5 | 1.1 | 0.72 | 221.9 | 154.0 | 5.1 | 81.1 | 0.0833 |
| | 10 | 1.2 | 0.69 | 230.7 | 227.6 | 13.4 | 87.6 | 0.1119 |
| | 20 | 1.0 | 0.67 | 205.1 | 286.1 | 16.2 | 90.1 | 0.1130 |
| | 50 | 1.1 | 0.68 | 195.4 | 343.1 | 17.3 | 91.7 | 0.1052 |
| **4** | 5 | 1.1 | 0.73 | 228.9 | 49.9 | 9.6 | 49.6 | 0.1447 |
| | 10 | 1.3 | 0.75 | 192.8 | 112.4 | 18.9 | 77.2 | 0.1672 |
| | 20 | 1.2 | 0.72 | 193.4 | 168.1 | 19.5 | 84.0 | 0.1415 |
| | 50 | 1.2 | 0.71 | 182.7 | 210.4 | 18.6 | 86.9 | 0.1366 |
| **5** | 5 | 11.5 | 0.74 | 247.3 | 35.4 | 36.8 | 58.4 | 0.0020 |
| | 10 | 8.0 | 0.88 | 214.8 | 68.1 | 16.7 | 64.6 | 0.0016 |
| | 20 | 8.5 | 0.95 | 170.8 | 265.7 | 5.4 | 88.9 | 0.0017 |
| | 50 | 6.3 | 0.62 | 89.9 | 492.8 | 6.7 | 94.0 | 0.0014 |
| **6** | 5 | 1.0 | 0.70 | 174.4 | 134.5 | 14.4 | 79.9 | 0.1245 |
| | 10 | 1.1 | 0.72 | 129.4 | 169.8 | 29.8 | 85.0 | 0.0893 |
| | 20 | 1.1 | 0.68 | 123.2 | 248.4 | 28.7 | 89.2 | 0.1065 |
| | 50 | 1.2 | 0.63 | 108.2 | 335.3 | 29.5 | 91.8 | 0.1065 |
| **7** | 5 | 1.5 | 0.70 | 172.7 | 201.8 | 5.1 | 85.5 | 0.0024 |
| | 10 | 1.0 | 0.65 | 153.4 | 224.1 | 0.5 | 86.6 | 0.0036 |
| | 20 | 0.9 | 0.63 | 158.1 | 272.4 | 2.6 | 89.1 | 0.0045 |
| | 50 | 1.0 | 0.60 | 153.2 | 328.2 | 2.9 | 90.9 | 0.0040 |

According to the shape of the semicircle, two time constants (using the circuit figure 3b) can be attributed, the charge transference resistance and the second to adsorbed molecules resistance [36–38]. The depressed semicircle is attributed to frequency dispersion effect and surface irregularities and heterogeneities [39].

It can be observed that the $Z_{real}$ value presents a large variation (table 1). Based on these results, it can be inferred that the presence of a halogen substituent in para position of the aromatic ring can modulate the inhibition capacity of the compound.

After obtaining the Nyquist diagrams of the compounds at different concentrations, the impedance results for API 5 L X52 steel with and without inhibitor can be explained by an equivalent circuit (figure 3) which comprises $R_{ct}$ (charge transfer resistance); Q, the constant phase element in parallel with $R_s$ (solution resistance); and $R_{mol}$, the molecules resistance.

Constant phase elements have widely been used [40] to account for deviations brought about by surface roughness. The impedance of CPE is given by the next equation [41]:

$$Z_{CPE} = Q^{-1}(j\omega)^{-n}, \tag{3.1}$$

where $Y_0$ is the magnitude of the CPE, n the CPE exponent (phase shift), $\omega$ the angular frequency ($\omega = 2\pi f$, where f is the AC frequency), and j here is the imaginary unit. The capacity correction to its real values was calculated from equation (3.2), where $\omega_{max}$ is the frequency at which the imaginary part of the impedance ($-Z_{imag}$) has a maximum. $C_{dl}$ represents the double-layer capacitance

$$C_{dl} = Y_0(\omega''_m)^{n-1}. \tag{3.2}$$

The value of inhibition efficiency ($\eta$) can be obtained by means of the following equation [42,43]:

$$\eta\,(\%) = \frac{(1/Rp)blank - (1/Rp)inhibitor}{(1/Rp)blank} \times 100. \tag{3.3}$$

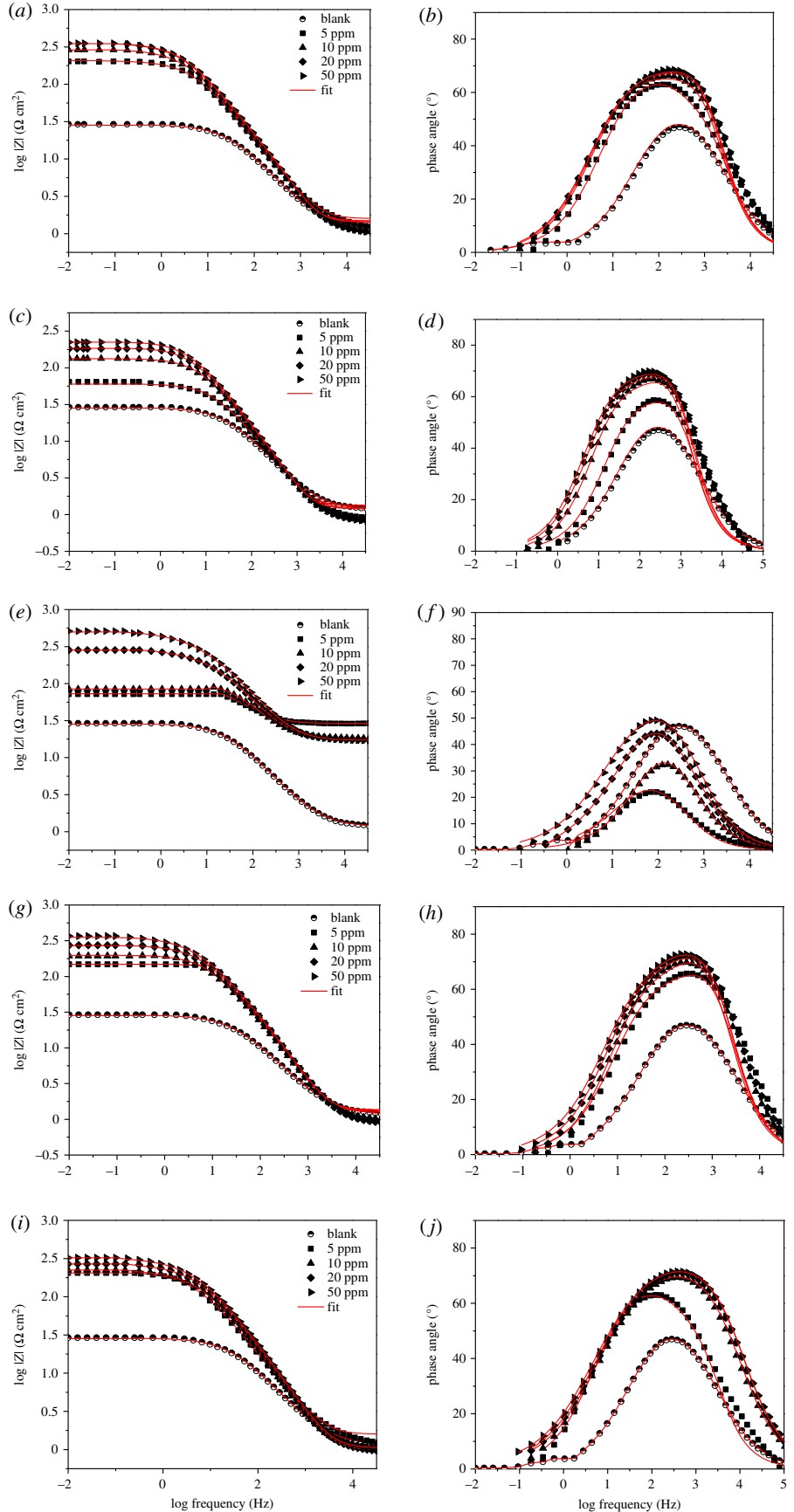

**Figure 5.** Bode diagrams of theophylline – triazoles compounds: (*a,b*) **3**, (*c,d*) **4**, (*e,f*) **5**, (*g,h*) **6** and (*i,j*) **7** for immersed API 5 L X52 steel at different inhibitor concentrations.

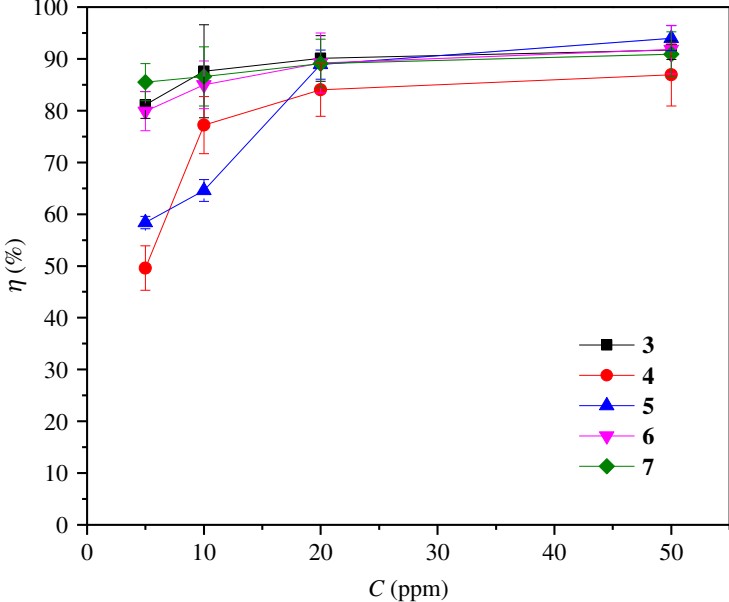

**Figure 6.** Variation of the inhibition efficiency of the theophylline–triazole derivatives **3**–**7** as a function of their concentration for API 5 L X70 steel submerged in 1 M HCl by EIS technique.

The Bode plots for API 5 L X52 steel in the absence and presence of various concentrations of 1,2,3-triazoles 1,4-disubstituted are represented in figure 5. An ideal capacitor is characterized by a fixed value of slope (unity) and phase angle ($-90°$). The increase in the values of phase angle in the presence of different inhibitor concentrations suggest that surface roughness of API 5 L X52 steel decreased due to the formation of protective film by theophylline–triazole derivative [44]. The result indicated that there were two coupled time constants, and the system could be described by three resistances which consist of electrolyte resistance ($R_s$), charge transfer resistance ($R_{ct}$), organic molecules adsorbed resistance ($R_{mol}$) and double-layer capacitance, as shown in figure 3b. While, in the absence of inhibitor, the phase angle versus log frequency show one time constant attributed to charge transfer resistance [45].

Finally, the corrosion inhibition effectiveness of 1,2,3-triazoles 1,4-disubstituted can also be interpreted in the real impedance values axis of the Bode modulus plots which also increase as the concentration increases (figure 5a,c,e,g,h).

In the electrochemical parameters obtained from the adjustment of experimental data with the equivalent circuit shown previously (table 1), it is possible to notice that the value of the capacitance of the electrochemical double layer ($C_{dl}$) decreases when more concentration of the inhibitor is added, due to the gradual displacement of water molecules with the theophylline–triazole inhibitor molecules in the working electrode, which decreases the number of active sites and consequently delays the corrosion phenomenon [46,47]. The charge transference resistance ($R_{ct}$) also increases when more concentration of the inhibitor is added in every case. Finally, the value of the inhibition efficiency reached a maximum of 94% of effectivity at 50 ppm for the compound containing chloride in its chemical structure.

In figure 6, a comparison of inhibition efficiency values for each theophylline–triazole derivative is shown. It is important to mention that, for the lowest concentration measurements (5 ppm), the best inhibition efficiency is presented by compound **7**.

The presence of the halogen attached to the benzene ring does not show a clear trend in the improvement of the inhibition efficiency, because it is not the only fragment that is interacting with the metal surface [48]. With respect to the highest inhibitor concentration (50 ppm), the best inhibition efficiency is presented by compound **5** ($\eta$ (%) = 94), followed closely by compound **3** and **6** ($\eta$ (%)∼91). The best inhibitor (compound **5**) can be attributed to the increased electron affinity of Cl, that could create a partially negative charge that can interact more efficiently with the positively charged metal surface, enhancing the overall adsorption of the molecule and, thus, increasing its corrosion inhibition efficiency.

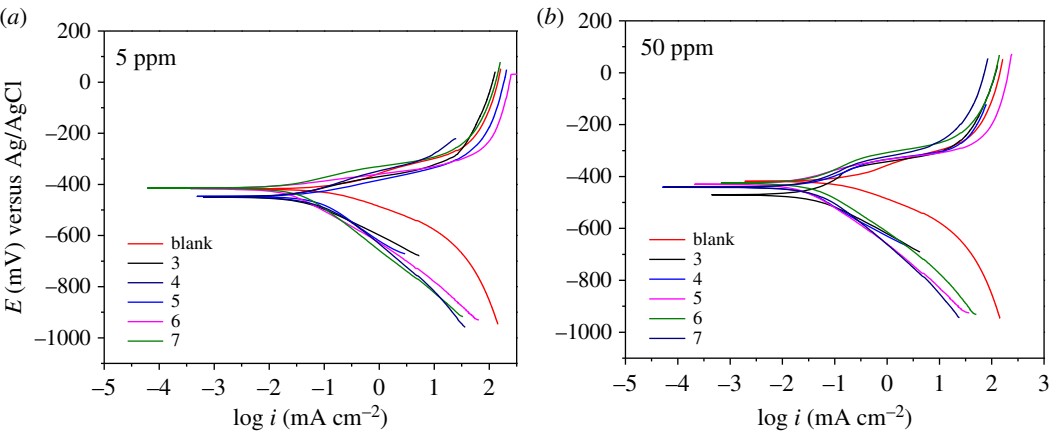

**Figure 7.** Potentiodynamic polarization curves of 1,2,3-triazoles 1,4-disubstituted in API 5 L X52 steel immersed in 1 M of HCl.

**Table 2.** Electrochemical parameters obtained by means of polarization curves for 1,2,3-triazoles 1,4-disubstituted in API 5 L X52 steel immersed in HCl 1M.

| inhibitor | C (ppm) | $E_{corr}$ (mV) versus Ag/AgCl | bc (mV dec-1) | ba (mV dec-1) | $I_{corr}$ (mA cm$^{-2}$) | $\eta$pol (%) |
|---|---|---|---|---|---|---|
| blank | 0 | −421.2 | −106.5 | 84.6 | 0.32 | — |
| 3 | 5 | −450.8 | −107.0 | 54.3 | 0.04 | 87.3 |
| 3 | 50 | −470.1 | −109.5 | 75.0 | 0.04 | 86.9 |
| 4 | 5 | −448.0 | −163.4 | 80.4 | 0.08 | 76.6 |
| 4 | 50 | −439.9 | −158.8 | 74.7 | 0.04 | 87.0 |
| 5 | 5 | −447.6 | −146.8 | 52.6 | 0.07 | 79.4 |
| 5 | 50 | −441.0 | −110.2 | 66.3 | 0.02 | 71.0 |
| 6 | 5 | −413.4 | −122.1 | 30.5 | 0.02 | 94.3 |
| 6 | 50 | −427.2 | −136.2 | 48.4 | 0.02 | 93.5 |
| 7 | 5 | −410.8 | −161.2 | 52.1 | 0.03 | 90.3 |
| 7 | 50 | −425.3 | −132.2 | 69.5 | 0.04 | 88.2 |

Also worth mentioning is the fact that the corrosion inhibition efficiency of all of the theophylline–triazole derivatives is higher than the corrosion inhibition observed in the underivatized theophylline [23] but is comparatively lower than other 1,2,3-triazole derivatives reported earlier [28–30].

## 3.2. Potentiodynamic polarization evaluation

The potentiodynamic polarization curves of the API 5 L X52 steel immersed in 1 M HCl in the absence and presence of triazoles 1,4-disubstituted (5 and 50 ppm) are shown in figure 7.

The parameters corrosion potential ($E_{corr}$), Tafel anodic pendant (ba), cathodic pendant (bc) and corrosion current density ($i_{corr}$) obtained from the curves are shown in table 2.

The inhibition efficiency ($\eta_{pol}$) is calculated with the following equation [49,50]:

$$\eta_{pol}(\%) = \frac{i_{corr\,blank} - i_{corr\,inhibitor}}{i_{corr\,blank}} \times 100, \tag{3.4}$$

where $i_{corr}$ is the corrosion current density in the absence and presence of the inhibitor.

Figure 7 shows that all of the curves move to lower current densities for both the anodic and cathodic half-reactions with the addition of 1,2,3-triazoles 1,4-disubstituted in API 5 L X52 steel immersed in 1 M HCl, and the trend is more pronounced at 5 ppm inhibitor concentration; indicating that the anodic dissolution of API 5 L X52 steel and cathodic reduction of hydrogen ions were inhibited [51].

This suggests that the rate of electrochemical reaction was reduced due to the formation of a protective layer of inhibitor molecules over the steel surface [52].

**Table 3.** Adjustment of thermodynamic data with the Langmuir isotherm.

| compound | $K_{ads}$ | $\Delta G^{\circ}_{ads}$ (kJ mol$^{-1}$) | slopes (M) | $R^2$ |
|---|---|---|---|---|
| **3** | $2.66 \times 10^7$ | $-41.7$ | $C/\theta = 1.0759C$ | 0.9982 |
| **4** | $5.95 \times 10^6$ | $-38.1$ | $C/\theta = 1.0788C$ | 0.9979 |
| **5** | $5.38 \times 10^6$ | $-37.8$ | $C/\theta = 0.9749C$ | 0.9965 |
| **6** | $2.66 \times 10^7$ | $-41.7$ | $C/\theta = 1.0700C$ | 0.9998 |
| **7** | $5.37 \times 10^7$ | $-43.4$ | $C/\theta = 1.0895C$ | 0.9989 |

Furthermore, the cathodic Tafel curves are parallel (figure 7a,b), which shows that there is no change in the hydrogen evolution mechanism with the addition of 1,2,3-triazoles 1,4-disubstituted and the reduction of hydrogen ions mainly takes place through a charge transfer [53].

In table 2, the electrochemical parameters are summarized. As can be seen, the corrosion current density decreases as the inhibition efficiency increases, which is attributed to the adsorption of the organic compound on the metal surface in 1 M HCl, generating a blockage of the active sites [54,55].

On the other hand, the corrosion potential ($E_{corr}$) is less than 85 mV, which suggests that it belongs to the mixed type with cathodic predominance for the two concentrations studied [56].

It is worth noting that the inhibition efficiency was also calculated by this technique for two concentrations of each organic compound (summarized in table 2), which closely correlate with the results obtained by the EIS technique.

## 3.3. Adsorption isotherm

Among all of the adsorption mechanism descriptions reported in the literature [57–60], the most common model to describe this process is Langmuir's isotherms (equation (3.5)). The corresponding adjustment for this model was performed, and the adjustment parameters are shown in table 3.

$$\frac{C}{\theta} = \frac{1}{K_{ads}} + C, \tag{3.5}$$

where $C$ is the concentration, $\theta$ is the coating degree and $K_{ads}$ is the adsorption constant.

The value of $K_{ads}$ is related with the Gibbs free energy value ($\Delta G^{\circ}_{ads}$) and is related to the following equation [61]:

$$\Delta G^{\circ}_{ads} = -RT\ln(55.5\,K_{ads}), \tag{3.6}$$

where the numeric value of 55.5 is the molar concentration of water in an acid solution, $R$ is the ideal gas constant and $T$ is the absolute temperature of the system.

The calculated values of the thermodynamic adjustment are also shown in table 3. Several authors mention that the values of $\Delta G^{\circ}_{ads}$ around $-40$ KJ mol$^{-1}$ or more negative are consistent with the charge interchange between the metal and the organic compound, so the reaction is defined as a chemisorption, while the values of $\Delta G^{\circ}_{ads}$ lower than $-20$ KJ mol$^{-1}$ produce only an electrostatic interaction (physisorption) [62,63].

According to these results, the compounds **3**, **6** and **7** present chemisorption process and, for the rest of the compounds, a physisorption–chemisorption type process is observed (figure 8) [64–66].

## 3.4. Adsorption mechanism

The corrosion inhibition of API 5 L X52 in 1 M HCl provoked by 1,2,3-triazoles 1,4-disubstituted molecules can be explained as follows (figure 9): The protonated species of the organic compounds interact, throughout Coulombic forces, with previously adsorbed chlorides ions (Cl$^-$) present on the API 5 L X52 steel surface, which results in physisorption of the inhibitor molecules. This way, the inhibitor molecules compete with H$^+$ for electrons on steel surfaces [53]. Moreover, the donation of lone electrons pairs of nitrogen atoms to the empty orbital of Fe atoms would induce a chemisorption process of the organic molecule [60] and the accumulation of negative charges on the steel surface can be transferred from the d orbital of Fe to unoccupied $\pi_*$ (anti-bonding) of 1,2,3-triazoles 1,4-disubstituted molecules (retro-donation) [67] (figure 9a,b). The combination of these two types of

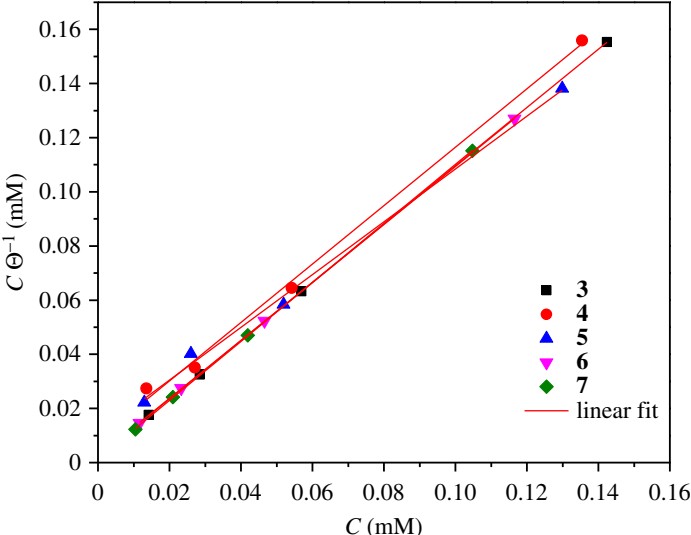

**Figure 8.** Adjustment of the thermodynamic analysis of the theophylline – triazole derivatives in API 5 L X52 steel immersed in 1 M HCl by using the Langmuir model.

| compound | R |
|----------|------|
| 3 | –H |
| 4 | –F |
| 5 | –Cl |
| 6 | –Br |
| 7 | –I |

(a) $R = Cl$    (b) $R = I$

---- chemical adsorption

---- physical adsorption

Cl⁻  Cl⁻  Cl⁻  Cl⁻  Cl⁻          Cl⁻  Cl⁻  Cl⁻  Cl⁻  Cl⁻

charged API 5 L X52 steel surface        charged API 5 L X52 steel surface

**Figure 9.** Schematic of the adsorption behaviour of inhibitor on API 5 L X52 steel surface immersed in 1 M HCl.

adsorption mechanisms is carried out for the compounds evaluated. However, the compound **7** (figure 9b) showed a stronger interaction (chemisorption) by nitrogen heteroatoms that have free electron pairs with steel, demonstrated with thermodynamic analysis using the Langmuir model.

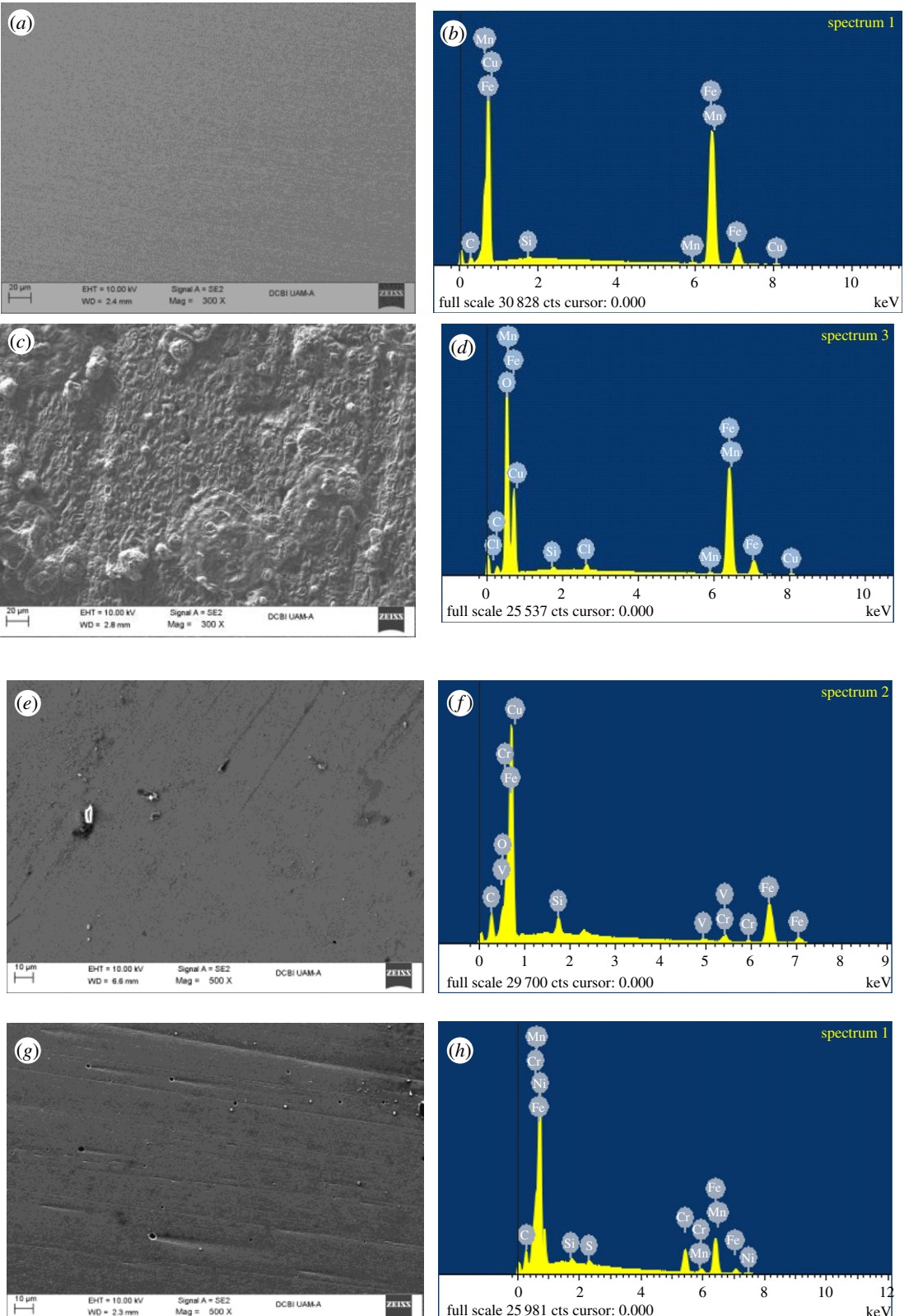

**Figure 10.** Images of SEM-EDS of API 5 L X52 steel for (*a,b*) polished steel, (*c,d*) immersed in 1 M HCl and in the presence of 50 ppm of (*e,f*) compound **5** and (*g,h*) compound **7**.

## 3.5. SEM-EDS surface analysis

The surfaces of the API 5 L X52 steel with and without inhibitor (figure 10) were characterized by SEM-EDS to corroborate the effectiveness of the inhibitor by evaluating the electrochemical response.

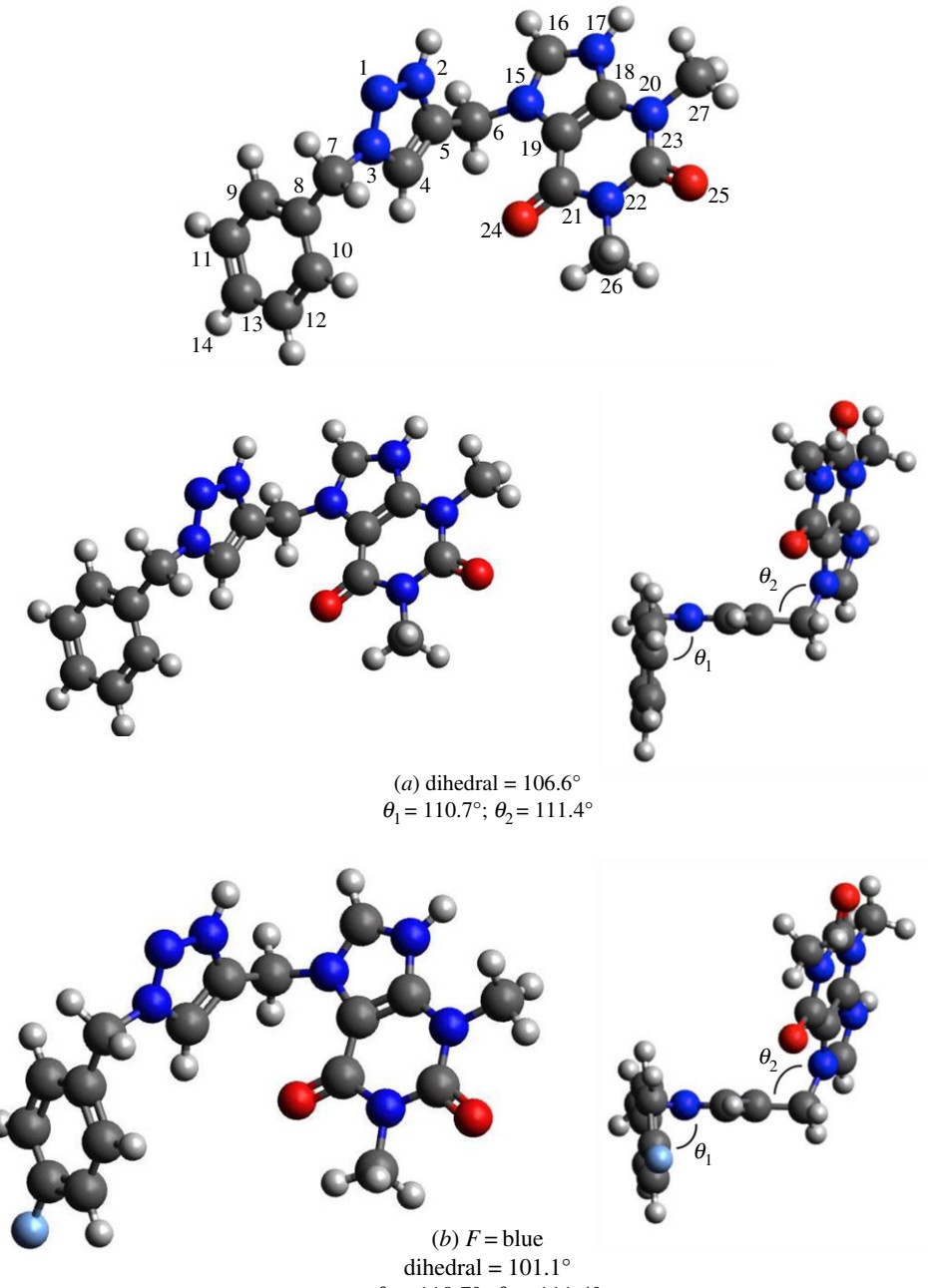

(a) dihedral = 106.6°
$\theta_1 = 110.7°$; $\theta_2 = 111.4°$

(b) $F$ = blue
dihedral = 101.1°
$\theta_1 = 110.7°$; $\theta_2 = 111.4°$

**Figure 11.** Optimized molecular structures: (a) compound **3**, (b) compound **4**, (c) compound **5**, (d) compound **6**, (e) compound **7**. Carbon atoms are represented by grey ball, Nitrogen by dark blue, Oxygen by red and Hydrogen by white colour. Dihedral angle among 3,7,8 and 10 atoms.

Figure 10a shows the surface of polished steel, while figure 10c shows the steel surface after 24 h of immersion in a solution 1 M HCl. As can be seen, the metallic surface presents damage due to the presence of chloride ions in the corrosive solution, supported by EDS and shown in figure 10d. Finally, figure 10e,g shows the morphology of the steel sample surface in the presence of the best organic inhibitors found in this research (compounds **5** and **7**). These results suggest that the inhibitors form protective films on the surface of the API 5 L X52 steel, effectively diminishing corrosion. In this case, the corrosive species ($Cl^-$) are not observed in the chemical analysis (figure 10f,h). However, as can be noted from figure 10 in the revised version the peak associated with carbon is much more intense in the presence of the inhibitor (see figure 10f,h) than on the bare steel sample (see figure 10b), therefore, this may indicate the presence of the inhibitors on the steel sample surface.

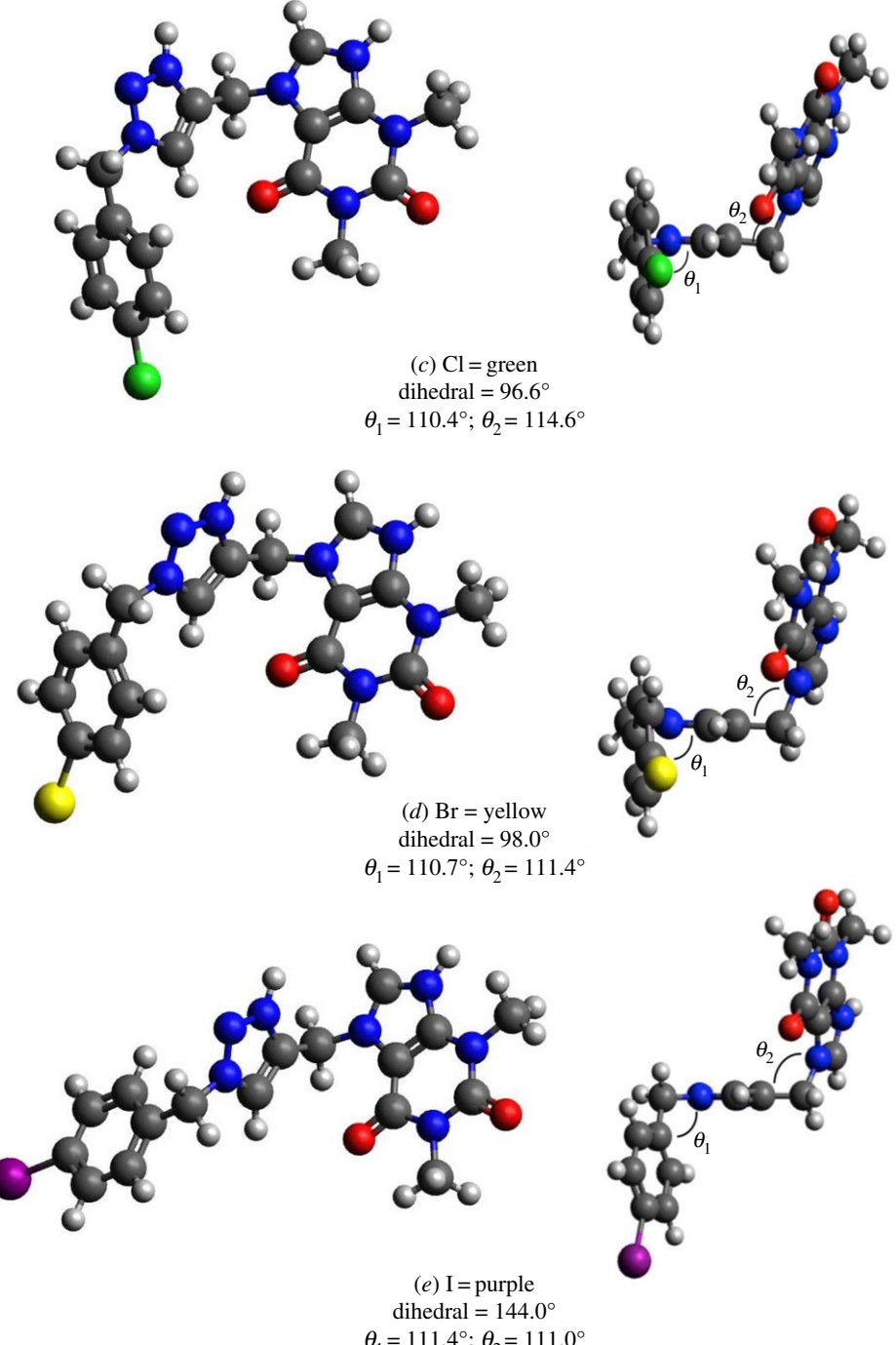

(*c*) Cl = green
dihedral = 96.6°
$\theta_1$ = 110.4°; $\theta_2$ = 114.6°

(*d*) Br = yellow
dihedral = 98.0°
$\theta_1$ = 110.7°; $\theta_2$ = 111.4°

(*e*) I = purple
dihedral = 144.0°
$\theta_1$ = 111.4°; $\theta_2$ = 111.0°

**Figure 11.** (*Continued.*)

## 3.6. Theoretical assessment

The presence of nitrogen as heteroatom in the inhibitor molecules provides high tendency toward protonation in aqueous acidic medium. Under these conditions, the calculations were carried out for the complete set of electrons and the geometry of involved structures was fully optimized. In other words, the nitrogen atoms hold positive charge by adding of hydrogen atoms and the solvent (water) effect is including with the SMD [33]. The results of the geometry optimization of the selected compounds in acidic medium and solvent effect are presented in figure 11.

A very important fact in the adsorption over the metallic surface is the planar configuration of the inhibitor molecules. The frameworks of these geometries show no planar configurations for all the

**Table 4.** Values for $E_{HOMO}$, $E_{LUMO}$, GAP, global hardness ($\eta$) for the inhibitors (*in vacuo*/with solvent) and total energy stabilization by solvent effect ($\Delta E_{solv}$).

| compound | $E_{HOMO}$ (eV) | $E_{LUMO}$ (eV) | GAP (eV) | $\eta$ (eV) | $\Delta E_{solv}$ (eV) |
|---|---|---|---|---|---|
| **3** | $-13.38/-8.59$ | $-7.90/-1.23$ | 5.48/7.36 | 2.74/3.68 | $-7.88$ |
| **4** | $-13.62/-8.64$ | $-7.97/-1.25$ | 5.64/7.39 | 2.82/3.70 | $-8.08$ |
| **5** | $-13.11/-8.50$ | $-7.96/-1.27$ | 5.15/7.23 | 2.58/3.61 | $-8.06$ |
| **6** | $-12.56/-8.33$ | $-7.94/-1.25$ | 4.61/7.08 | 2.31/3.54 | $-8.05$ |
| **7** | $-11.94/-7.78$ | $-7.92/-1.27$ | 4.03/6.52 | 2.01/3.26 | $-8.02$ |

**Table 5.** Values for chemical potential and electrophilicity (*in vacuo*/with solvent).

| compound | chemical potential ($\mu$) (eV) | electrophilicity (W) (eV) |
|---|---|---|
| **3** | $-10.64/-4.91$ | 20.65/3.27 |
| **4** | $-10.80/-4.94$ | 20.65/3.31 |
| **5** | $-10.53/-4.89$ | 21.53/3.30 |
| **6** | $-10.25/-4.79$ | 22.76/3.24 |
| **7** | $-9.93/-4.52$ | 24.48/3.14 |

inhibitors. Some angles are shown in figure 11, the halogen substitute has an important effect over the dihedral angle, particularly with iodine.

Quantum chemical indices $E_{HOMO}$, $E_{LUMO}$, GAP, hardness $\eta$ and dipole moment are summarized in table 4. It is known that $E_{HOMO}$ is often related to electron donation ability of the inhibitor molecules toward the metallic surface atoms, and it is expected that a higher $E_{HOMO}$ value would favour a greater charge transfer [68]. According to these results, the values of $E_{HOMO}$ show the following behaviour: compound **7** > compound **6** > compound **5** > compound **3** > compound **4**. In this case, the notorious largest $E_{HOMO}$ corresponds to compound **7** in line with the aforementioned experiments at low concentrations and the lowest $E_{HOMO}$ corresponds to compound **4** that in terms of activity is the worst inhibitor.

Some others authors (for instance [69]) have found that a smaller value of the hardness is related to greater stability of the surface-inhibitor complex formed. The trend that we got was **4** > **3** > **5** > **6** > **7**, as can be observed in table 4. Then the complex metallic surface-compound **7** presents the best stability and inhibition efficiency at low concentrations.

Moreover, smaller values of $E_{LUMO}$ present better capacity of the inhibitor to accept electrons, for this property the trend with solvent was compound **7**, **5** < **6**, **4** < **3**. Negative values of $E_{LUMO}$ indicate the capability to accept electrons too. Nevertheless, there is not a clear relationship between the above last two properties.

For the overall electrophilicity and according to the computed values (table 5), compound **7** exhibited a better nucleophilic character with solvent effect. Whereas compound **4** was lower than the rest. Therefore, a better interaction of compound **7** with the metallic surface would be expected than for the rest of compounds, the differences are not significant.

The Hirshfeld atomic charges for the most important centres with solvent effect are summarized in table 6. Redistribution of the charge is observed by the presence of the substituent, the atoms that have a larger negative charge, suggesting that those are active centres with excess charges that could act as a nucleophilic group. The most favourable sites for the interaction with the metal surface are the oxygen atoms, 24 and 25 (see figure 11 and table 6), for all the selected compounds. As it has been expected, fluorine in compound **4** is the best electro-attractor followed by chlorine; the halogen substitutes are active sites for the inhibition. The carbon 8, in all the compounds, is another nucleophilic centre. The carbon 13 in compound **7** also gains charge.

**Table 6.** Atomic Hirshfeld charges for the different inhibitor compounds. See figure 11 for the atomic labels.

| no. | atom | compound **3** | compound **4** | compound **5** | compound **6** | compound **7** |
|---|---|---|---|---|---|---|
| 1 | N | −0.010 | −0.011 | −0.005 | −0.012 | −0.003 |
| 2 | N | 0.303 | 0.307 | 0.307 | 0.308 | 0.307 |
| 3 | N | 0.088 | 0.087 | 0.087 | 0.086 | 0.090 |
| 4 | C | 0.171 | 0.161 | 0.171 | 0.157 | 0.183 |
| 5 | C | 0.074 | 0.074 | 0.073 | 0.074 | 0.075 |
| 6 | C | 0.273 | 0.272 | 0.276 | 0.272 | 0.273 |
| 7 | C | 0.210 | 0.215 | 0.222 | 0.216 | 0.220 |
| 8 | C | −0.020 | −0.015 | −0.013 | −0.012 | −0.013 |
| 9 | C | 0.018 | 0.048 | 0.049 | 0.047 | 0.036 |
| 10 | C | 0.026 | 0.052 | 0.030 | 0.049 | 0.013 |
| 11 | C | 0.015 | 0.028 | 0.027 | 0.013 | 0.029 |
| 12 | C | 0.015 | 0.028 | 0.029 | 0.013 | 0.028 |
| 13 | C | 0.016 | 0.096 | 0.039 | −0.004 | −0.029 |
| 14 | H | 0.000 | −0.166(F) | −0.090(Cl) | −0.037(Br) | −0.035(I) |
| 15 | N | 0.024 | 0.024 | 0.023 | 0.024 | 0.024 |
| 16 | C | 0.356 | 0.354 | 0.355 | 0.354 | 0.356 |
| 17 | N | 0.235 | 0.234 | 0.234 | 0.234 | 0.235 |
| 18 | C | 0.154 | 0.154 | 0.154 | 0.154 | 0.154 |
| 19 | C | 0.012 | 0.012 | 0.011 | 0.013 | 0.012 |
| 20 | N | −0.030 | −0.030 | −0.030 | −0.030 | −0.030 |
| 21 | C | 0.204 | 0.205 | 0.203 | 0.205 | 0.204 |
| 22 | N | −0.040 | −0.040 | −0.040 | −0.040 | −0.040 |
| 23 | C | 0.264 | 0.264 | 0.264 | 0.264 | 0.264 |
| 24 | O | −0.348 | −0.343 | −0.334 | −0.338 | −0.347 |
| 25 | O | −0.380 | −0.381 | −0.380 | −0.381 | −0.380 |
| 26 | C | 0.213 | 0.212 | 0.213 | 0.213 | 0.160 |
| 27 | C | 0.159 | 0.159 | 0.126 | 0.159 | 0.213 |

Experimental data reveal that compound **7** is the best inhibitor of the corrosion, the geometric structure and the parameters here studied such as $E_{HOMO}$, hardness, electrophilicity and atomic charges support this fact.

## 4. Conclusion

The theophylline–triazole derivatives synthesized by heterogeneous catalysis were obtained in good yields. These compounds were evaluated as corrosion inhibitors in API 5 L X52 steel, demonstrating that the inhibition activity of the theophylline–triazole derivatives is greater than the inhibition activities of any xanthine derivatives, such as theophylline and theobromine.

The best inhibitors are the ones bearing a chlorine (compound **5**) and iodine (compound **7**) atoms at the para position of the aromatic ring. The best inhibition activity is reached at a concentration of 50 ppm ($\eta$ of 87%). Compound **4** is the least efficient as a corrosion inhibitor, containing halogen fluoride.

These facts were confirmed by theoretical results; compound **7** exhibits a better nucleophilic character; the highest $E_{HOMO}$ hence underlining its good ability as an electron donor; it also has the largest hardness with solvent effect. On the other hand, the worst inhibitor is the compound **4**.

The adsorption study for compounds **3**–**7** showed that the corrosion inhibition process follows the Langmuir isotherm, with a combined physisorption–chemisorption process for compounds **3**, **6** and **7**. Finally, the inhibitors that carried out chemisorption process are compounds **4** and **5**.

The negative values of $E_{HOMO}$ corroborate the physical adsorption of the inhibitors tested. The atomic charges give the sites atoms with high probability where the interaction with the metallic surface could be. In the whole set of compounds here studied, the oxygen atoms have a large electron contribution; they can performance as electron-donating atoms. Compound **7** has one additional negative charged atom, C13.

Data accessibility. Data available from the Dryad Digital Repository: https://doi.org/10.5061/dryad.k9v03g0 [70].
Authors' contributions. A.E.-V. carried out the experiments, analysis and interpretation of data by electrochemical evaluation; D.Á.-B. performed the surface characterization. F.J.R.-G. and M.P.-P. supervised all the electrochemical experiments. G.E.N.-S., L.L.R. and D.P.-M. supervised the synthesis and characterization of all the triazole–theophylline compounds. I.K.M.-C. was responsible for the synthesis and purification of organic compounds. A.M.N.-L. performed the quantum chemical calculations. All the authors gave their final approval for publication.
Competing interests. We declare we have no competing interests.
Funding. The authors thank CONACyT for the financial support granted for the development of this research through the project no. P43984-Q. A.M.N.-L. was partially supported by PRODEP program.
Acknowledgements. The authors thank the Laboratorio de Microscopia Divisional of the Universidad Autónoma Metropolitana-Azcapotzalco for the use of the Scanning Electron Microscope SUPRA 55 VP and the Laboratorio de Supercómputo y Visualización en Paralelo at the Universidad Autónoma Metropolitana-Iztapalapa for access to their computer facilities. A.E.-V. and F.J.R.-G. express their gratitude to the Facultad de Química (UNAM), Departamento de Ingeniería Metalurgica, and CONACyT for providing a postdoctoral fellowship.

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
