## [Reviewer comments · Royal Society Open Science]

Review History

RSOS-181738.R0 (Original submission)

Review form: Reviewer 1 (Moses Solomon)

Is the manuscript scientifically sound in its present form?

Yes

Are the interpretations and conclusions justified by the results?

Yes

Is the language acceptable?

Yes

Is it clear how to access all supporting data?

Yes

Do you have any ethical concerns with this paper?

No

Have you any concerns about statistical analyses in this paper?

No

Recommendation?

Accept with minor revision (please list in comments)

Comments to the Author(s)

Comments on RSOS-181738

Title: Adsorption behavior of new theophylline-triazole based derivatives as effective corrosion inhibitors for steel in acidic medium

Series of theophylline derivatives containing 1,2,3-triazole moieties were synthesized, characterized, and studied as corrosion inhibitor for API 5L X52 steel in 1 M HCl medium by Espinoza-Vázquez et al. Electrochemical techniques namely EIS and PDP were used in the investigation and was supported with SEM and EDS analysis. The study is of interest and the results presented well interpreted. However, MINOR revision needs to be done before publication. Here are some of the issues:

1. Individual graph in the PDP plot should be labeled since authors are using line plotting. Otherwise, they should replot with line and symbol.
2. The mechanism of adsorption of the studied inhibitor should be fully discussed.
3. The authors should present the OCP-time graph to convince readers that OCP stability was achieved before measurements.
4. Additional surface analysis like AFM, XRD, and/or XPS is needed. This will greatly improve the quality of this manuscript.

Review form: Reviewer 2 (Vandana Srivastava)

Is the manuscript scientifically sound in its present form?

Yes

Are the interpretations and conclusions justified by the results?

Yes

Is the language acceptable?

Yes

Is it clear how to access all supporting data?

Yes

Do you have any ethical concerns with this paper?

No

Have you any concerns about statistical analyses in this paper?

No

Recommendation?

Accept with minor revision (please list in comments)

Comments to the Author(s)

The authors have studied five new Theophylline derivatives bearing Triazole moieties for their corrosion inhibition behavior on steel in 1M HCl solution. All the synthesized inhibitors have been very well characterized through their spectral data. The paper can be considered for publication after satisfactory response to following comments:

1. How did the authors convert Molar solution to ppm concentration? Did the authors check the inhibiting effect of DMF alone and in the presence of inhibitor because DMF also acts as inhibitor?
2. In EDX Fig, the presence of inhibitor is not visible. How did authors confirm the adsorption of inhibitor on steel surface?
3. There is no explanation of β_a and β_c values in Tafel polarization study. The sign of β_c should be negative.
4. The range for polarization ± 500 mV is very high. Is there any reason for this? The sweep rate should be 0.1mV /sec.
5. In EIS study, Bode plot should be provided to understand the adsorption behavior of inhibitors.
6. What is r_{mol} ? Is it film resistance? Clarify
7. The mechanism of inhibition should be clearly discussed. Which form of the inhibitor is predominantly adsorbed: Neutral or protonated? It should be clearly stated based on DFT calculation.
8. Adsorption isotherm should be redrawn by selecting proper values on X and Y axis
9. Fig 8: EDX is reported for 5 and 20 ppm concentration while optimum concentration is 50 ppm. Is there any reason for using lower concentration?
10. What is the significance of using API5LX52 and 1M HCl?

Review form: Reviewer 3 (Saviour Umoren)

Is the manuscript scientifically sound in its present form?

No

Are the interpretations and conclusions justified by the results?

No

Is the language acceptable?

No

Is it clear how to access all supporting data?

Yes

Do you have any ethical concerns with this paper?

No

Have you any concerns about statistical analyses in this paper?

No

Recommendation?

Major revision is needed (please make suggestions in comments)

Comments to the Author(s)

COMMENTS ON MANUSCRIPT RSOS-181738

TITLE: Adsorption behavior of new theophylline-triazole based derivatives as effective corrosion inhibitors for steel in acidic medium

COMMENTS

The manuscript reports on the synthesis, characterization of theophylline-triazole based derivatives and evaluation of corrosion inhibition performance for API 5L X52 steel corrosion in 1 M HCl solution. The corrosion inhibition effect was investigated using electrochemical impedance spectroscopy and potentiodynamic polarization techniques complemented by surface morphological characterization of the corroded carbon steel samples without and with the inhibitors with SEM/EDS. The manuscript is of interest in the fields of materials and corrosion. The synthesis and characterization of the compounds used as inhibitors is well treated but the same cannot be said of the corrosion inhibition behavior of the synthesized compounds. In its present form, the manuscript is not recommended for publication in Royal Society Open Science. Major revision of the paper is needed before consideration for publication. The issues to be addressed by the authors are appended below:

- (1) The Title of the manuscript should be modified to read "Adsorption and corrosion inhibition behavior of new theophylline-triazole based derivatives for steel in acidic medium"
- (2) There are important keywords not listed and the irrelevant ones are listed. Words like Acid, Theophylline, Corrosion inhibition, Steel should form part of the keywords if required by the journal.
- (3) On page 2 line 14, 'adsorpted' should be corrected to 'adsorbed'. Also in line 24, 'absord' should be corrected to 'adsorb'
- (4) The idea that the synthesized compounds were tested as 'environmentally friendly' is rather speculative and not based on experimental evidence as they were not tested experimentally to ensure that they meet the three criteria of a compound to be designated as 'environmentally friendly' which are toxicity, biodegradability and bioaccumulation. Hence the statement in line 31 of page 2 should be revised to clear this ambiguity.
- (5) It is not clear if the authors had check the purity of the synthesized compounds given that the melting point range is large as could be seen as follows: Compound 2 (220-226 oC); Compound 3 (169-172 oC), Compound 4 (182-186 oC); Compound 6 (194-198 °C.) Compound 7 (180-184 °C.).
- (6) The discussion on EIS measurements results should be rewritten. It is very shallow and should be expanded to convey meaning. Fundamental questions like the need of using constant phase element in the equivalent circuit need to be answered. Also for the purpose of clarity, the authors need to use appropriate scientific words to describe the technique. It is not clear what the authors meant by "the results were adjusted using equivalent electric circuits"
- (7) The authors should also present the EIS results in Bode formats as the number of time constants is better revealed in Bode representation than in Nyquist representation.
- (8) Again the discussion on potentiodynamic polarization technique should be rewritten as it is very shallow. The authors have not discuss the Fig.6 at all. The scope of the discussion should be expanded to make meaning.
- (9) In Table 1, the authors should replace the SD with Chi squared values (values of goodness of fit) to ascertain the validity of the equivalent circuit used to fit the experimental data.
- (10) The units of bc and ba are not correct. The units should be mV/dec
- (11) In Table 3, the authors should the replace linear regression equation with values of the slopes. Also values of Kads and not ln Kads should be given.
- (12) Finally the English language of this manuscript is very poor and need serious improvement. The assistance of native English speakers may be sought in this regards.

Dr. Saviour Umoren

Centre of Research Excellence in Corrosion

King Fahd University of Petroleum and Minerals
Dhahran, Saudi Arabia.

Decision letter (RSOS-181738.R0)

19-Nov-2018

Dear Dr Negrón Silva:

Title: Adsorption behavior of new theophylline-triazole based derivatives as effective corrosion inhibitors for steel in acidic medium

Manuscript ID: RSOS-181738

Thank you for submitting the above manuscript to Royal Society Open Science. On behalf of the Editors and the Royal Society of Chemistry, I am pleased to inform you that your manuscript will be accepted for publication in Royal Society Open Science subject to minor revision in accordance with the referee suggestions. Please find the reviewers' comments at the end of this email.

The reviewers and handling editors have recommended publication, but also suggest some minor revisions to your manuscript. Therefore, I invite you to respond to the comments and revise your manuscript.

Please also include the following statements alongside the other end statements. As we cannot publish your manuscript without these end statements included, if you feel that a given heading is not relevant to your paper, please nevertheless include the heading and explicitly state that it is not relevant to your work. We have included a screenshot example of the end statements for reference.

- Ethics statement

Please clarify whether you received ethical approval from a local ethics committee to carry out your study. If so please include details of this, including the name of the committee that gave consent in a Research Ethics section after your main text. Please also clarify whether you received informed consent for the participants to participate in the study and state this in your Research Ethics section.

OR

Please clarify whether you obtained the necessary licences and approvals from your institutional animal ethics committee before conducting your research. Please provide details of these licences and approvals in an Animal Ethics section after your main text.

OR

Please clarify whether you obtained the appropriate permissions and licences to conduct the fieldwork detailed in your study. Please provide details of these in your methods section.

- Funding statement

Please include a funding section after your main text which lists the source of funding for each author.

Because the schedule for publication is very tight, it is a condition of publication that you submit the revised version of your manuscript before 28-Nov-2018. Please note that the revision deadline will expire at 00.00am on this date. If you do not think you will be able to meet this date please let me know immediately.

Best wishes,
Dr Laura Smith
Publishing Editor, Journals

On behalf of the Subject Editor Professor Anthony Stace and the Associate Editor Dr Andrew Harned.

RSC Associate Editor:

Comments to the Author:

The referees raise several valid and important points that should be addressed by the authors.

RSC Subject Editor:

Comments to the Author:

(There are no comments.)

Reviewer comments to Author:

Reviewer: 1

Comments to the Author(s)

Comments on RSOS-181738

Title: Adsorption behavior of new theophylline-triazole based derivatives as effective corrosion inhibitors for steel in acidic medium

Series of theophylline derivatives containing 1,2,3-triazole moieties were synthesized, characterized, and studied as corrosion inhibitor for API 5L X52 steel in 1 M HCl medium by Espinoza-Vázquez et al. Electrochemical techniques namely EIS and PDP were used in the investigation and was supported with SEM and EDS analysis. The study is of interest and the results presented well interpreted. However, MINOR revision needs to be done before publication. Here are some of the issues:

1. Individual graph in the PDP plot should be labeled since authors are using line plotting. Otherwise, they should replot with line and symbol.
2. The mechanism of adsorption of the studied inhibitor should be fully discussed.
3. The authors should present the OCP-time graph to convince readers that OCP stability was achieved before measurements.
4. Additional surface analysis like AFM, XRD, and/or XPS is needed. This will greatly improve the quality of this manuscript.

Reviewer: 2

Comments to the Author(s)

The authors have studied five new Theophylline derivatives bearing Triazole moieties for their corrosion inhibition behavior on steel in 1M HCl solution. All the synthesized inhibitors have been very well characterized through their spectral data. The paper can be considered for publication after satisfactory response to following comments:

1. How did the authors convert Molar solution to ppm concentration? Did the authors check the inhibiting effect of DMF alone and in the presence of inhibitor because DMF also acts as inhibitor?
2. In EDX Fig, the presence of inhibitor is not visible. How did authors confirm the adsorption of inhibitor on steel surface?
3. There is no explanation of β_a and β_c values in Tafel polarization study. The sign of β_c should be negative.

4. The range for polarization ± 500 mV is very high. Is there any reason for this? The sweep rate should be 0.1mV /sec.
5. In EIS study, Bode plot should be provided to understand the adsorption behavior of inhibitors.
6. What is r_{mol} ? Is it film resistance? Clarify
7. The mechanism of inhibition should be clearly discussed. Which form of the inhibitor is predominantly adsorbed: Neutral or protonated? It should be clearly stated based on DFT calculation.
8. Adsorption isotherm should be redrawn by selecting proper values on X and Y axis
9. Fig 8: EDX is reported for 5 and 20 ppm concentration while optimum concentration is 50 ppm. Is there any reason for using lower concentration?
10. What is the significance of using API5LX52 and 1M HCl?

Reviewer: 3

Comments to the Author(s)

COMMENTS ON MANUSCRIPT RSOS-181738

TITLE: Adsorption behavior of new theophylline-triazole based derivatives as effective corrosion inhibitors for steel in acidic medium

COMMENTS

The manuscript reports on the synthesis, characterization of theophylline-triazole based derivatives and evaluation of corrosion inhibition performance for API 5L X52 steel corrosion in 1 M HCl solution. The corrosion inhibition effect was investigated using electrochemical impedance spectroscopy and potentiodynamic polarization techniques complemented by surface morphological characterization of the corroded carbon steel samples without and with the inhibitors with SEM/EDS. The manuscript is of interest in the fields of materials and corrosion. The synthesis and characterization of the compounds used as inhibitors is well treated but the same cannot be said of the corrosion inhibition behavior of the synthesized compounds. In its present form, the manuscript is not recommended for publication in Royal Society Open Science. Major revision of the paper is needed before consideration for publication. The issues to be addressed by the authors are appended below:

- (1) The Title of the manuscript should be modified to read "Adsorption and corrosion inhibition behavior of new theophylline-triazole based derivatives for steel in acidic medium"
- (2) There are important keywords not listed and the irrelevant ones are listed. Words like Acid, Theophylline, Corrosion inhibition, Steel should form part of the keywords if required by the journal.
- (3) On page 2 line 14, 'adsorpted' should be corrected to 'adsorbed'. Also in line 24, 'absord' should be corrected to 'adsorb'
- (4) The idea that the synthesized compounds were tested as 'environmentally friendly' is rather speculative and not based on experimental evidence as they were not tested experimentally to ensure that they meet the three criteria of a compound to be designated as 'environmentally friendly' which are toxicity, biodegradability and bioaccumulation. Hence the statement in line 31 of page 2 should be revised to clear this ambiguity.
- (5) It is not clear if the authors had checked the purity of the synthesized compounds given that the melting point range is large as could be seen as follows: Compound 2 (220-226 oC); Compound 3 (169-172 oC), Compound 4 (182-186 oC); Compound 6 (194-198 °C.) Compound 7 (180-184 °C.).
- (6) The discussion on EIS measurements results should be rewritten. It is very shallow and should be expanded to convey meaning. Fundamental questions like the need of using constant phase element in the equivalent circuit need to be answered. Also for the purpose of clarity, the

authors need to use appropriate scientific words to describe the technique. It is not clear what the authors meant by "the results were adjusted using equivalent electric circuits"

(7) The authors should also present the EIS results in Bode formats as the number of time constants is better revealed in Bode representation than in Nyquist representation.

(8) Again the discussion on potentiodynamic polarization technique should be rewritten as it is very shallow. The authors have not discuss the Fig.6 at all. The scope of the discussion should be expanded to make meaning.

(9) In Table 1, the authors should replace the SD with Chi squared values (values of goodness of fit) to ascertain the validity of the equivalent circuit used to fit the experimental data.

(10) The units of bc and ba are not correct. The units should be mV/dec

(11) In Table 3, the authors should the replace linear regression equation with values of the slopes. Also values of Kads and not ln Kads should be given.

(12) Finally the English language of this manuscript is very poor and need serious improvement. The assistance of native English speakers may be sought in this regards.

Dr. Saviour Umoren
Centre of Research Excellence in Corrosion
King Fahd University of Petroleum and Minerals
Dhahran, Saudi Arabia.

Author's Response to Decision Letter for (RSOS-181738.R0)

See Appendix A.

Decision letter (RSOS-181738.R1)

21-Jan-2019

Dear Dr Negrón Silva:

Title: Adsorption and corrosion inhibition behavior of new theophylline-triazole based derivatives for steel in acidic medium

Manuscript ID: RSOS-181738.R1

It is a pleasure to accept your manuscript in its current form for publication in Royal Society Open Science. The chemistry content of Royal Society Open Science is published in collaboration with the Royal Society of Chemistry.

On behalf of the Subject Editor Professor Anthony Stace and the Associate Editor Dr Andrew Harned.

RSC Associate Editor

Comments to the Author:

The authors have done a good job of responding to the concerns and questions raised by the previous review.

Reviewer(s)' Comments to Author:

Appendix A

January 11, 2019

Andrew Harned

Associate Editor, Royal Society Open Science
Texas Tech University

It is a pleasure to send the corrected version of the manuscript ID: RSOS-181738

Adsorption and corrosion inhibition behavior of new theophylline-triazole based derivatives for steel in acidic medium

Authors:

Araceli Espinoza-Vázquez^a, Francisco Javier Rodríguez-Gómez^a, Ivonne Karina Martínez-Cruz^b, Deyanira Ángeles-Beltrán,^b Guillermo E. Negrón-Silva^{b*}, Manuel Palomar-Pardavé,^c Leticia Lomas Romero,^c Diego Pérez-Martínez^c, Alejandra M. Navarrete-López^b

We have answered all comments and suggestions from the referees. We would be grateful for the attention given to our letter and accompanying manuscript and be happy to provide any further information as deemed necessary.

With our best regards,

Dr. Guillermo E. Negrón Silva

Profesor-Investigador

Laboratorio de Química de Materiales

Departamento de Ciencias Básicas

División de Ciencias Básicas e Ingeniería

Laboratorios G-112

Tel.: 53189593

Reviewer comments to Author:

Reviewer: 1

Comments to the Author(s)

Comments on RSOS-181738

Title: Adsorption behavior of new theophylline-triazole based derivatives as effective corrosion inhibitors for steel in acidic medium

Series of theophylline derivatives containing 1,2,3-triazole moieties were synthesized, characterized, and studied as corrosion inhibitor for API 5L X52 steel in 1 M HCl medium by Espinoza-Vázquez et al. Electrochemical techniques namely EIS and PDP were used in the investigation and was supported with SEM and EDS analysis. The study is of interest and the results presented well interpreted. However, MINOR revision needs to be done before publication. Here are some of the issues:

1. Individual graph in the PDP plot should be labeled since authors are using line plotting. Otherwise, they should replot with line and symbol

Reply: We agree with these comments; thus, we have modified Figure 3 as follows:

2. The mechanism of adsorption of the studied inhibitor should be fully discussed.

Reply: We agree with these comments, therefore, in the revised version we have added the following section regarding this comment:

Adsorption mechanism

The corrosion inhibition of API 5L X52 in HCl 1M provoked by 1,2,3-triazoles 1,4-disubstituted molecules can be explained as follows (Figure 9): The protonated species of the organic compounds interact, throughout Columbic forces, with previously adsorbed chlorides ions (Cl^-) present on the API 5L X52 steel surface which results in physisorption of the inhibitor molecules. This way, the inhibitor molecules compete with H^+ for electrons on steel surfaces [67]. Moreover, the donation of lone electrons pairs of nitrogen atoms to the empty orbital of Fe atoms would induce a chemisorption process of the organic molecule [68] and the accumulation of negative charges on the steel surface can be transferred from the d orbital of Fe to unoccupied π^* (anti-bonding) of 1,2,3-triazoles 1,4-disubstituted molecules (retro-donation) [69], see Figure 9a and 9b. The combination of these two types of adsorption mechanisms is carried out for the compounds evaluated. However, the compound 7 (Figure 9b) showed a stronger interaction

(chemisorption) by nitrogen heteroatoms that have free electron pairs with steel, demonstrated with thermodynamic analysis using the Langmuir model.

67. Hamania H, Douadia T, Daouda D, Al-Noaimic M, Rikkouha R, Chafaa S. 2017. 1-(4-Nitrophenyl-imino)-1-(phenylhydrazono)-propan-2-one as corrosion inhibitor for mild steel in 1 M HCl solution: Weight loss, electrochemical, thermodynamic and quantum chemical studies, *J. Electroanalyt. Chem.* 801,425–438.
68. Singh P, Quraishi MA. 2016. Corrosion inhibition of mild steel using Novel Bis Schiff's Bases as corrosion inhibitors: *Electrochemical and Surface. Measurement* 86, 114–124.
69. Haldhar R, Prasad D, Saxena A. 2018. Myristica fragrans extract as an eco-friendly corrosion inhibitor for mild steel in 0.5 M H₂SO₄ solution, *J. Environm. Chem. Eng.* 6, 2290–2301

Compound	R
3	-H
4	-F
5	-Cl
6	-Br
7	-I

Figure 9. Schematic representation of the adsorption behavior of inhibitor on API 5L X52 steel surface immersed in 1M HCl.

70. The authors should present the OCP-time graph to convince readers that OCP stability was achieved before measurements.

Reply: We agree with this comment therefore in the revised version we have included the following:

... Fig. 3a shows the variations in time of the open circuit potential (OCP) in static conditions at 50 ppm, at the electrode made of API 5L X52 steel in presence of different inhibitor concentrations. It should be mentioned that it was necessary the stabilization of the OCP previously to do the EIS determinations. The steady state was reached after 1700 seconds...

Figure 3 a) Open Circuit Potential (OCP) vs time of theophylline-triazoles derivatives at 50 ppm

71. Additional surface analysis like AFM, XRD, and/or XPS is needed. This will greatly improve the quality of this manuscript.

Reply: The reviewer is right, however, at this moment we do not have the equipment nor the resources to perform the requested surfaces analysis. Notwithstanding, we believe that the experimental results showed in the SEM-EDS section are sufficient to directly relate them to the anticorrosive capacity of the inhibitors reported in this work.

Reviewer: 2

Comments to the Author(s)

The authors have studied five new Theophylline derivatives bearing Triazole moieties for their corrosion inhibition behavior on steel in 1M HCl solution. All the synthesized inhibitors have been very well characterized through their spectral data. The paper can be considered for publication after satisfactory response to following comments:

1. How did the authors convert Molar solution to ppm concentration?

Reply: There was an initial concentrated solution in order to make dilutions and finally, diluted solutions were obtained by sampling different aliquots completing with DMF. However, inhibitor concentrations are usually reported as ppm (part per million, mg/kg or mg/liter).

Did the authors check the inhibiting effect of DMF alone and in the presence of inhibitor because DMF also acts as inhibitor?

Reply: In this case, the compounds have solubility in DMF, so it was used as a solvent, however, it was found that, at the evaluated working conditions, DMF corrosion inhibitor capacity is practically null.

2. In EDX Fig, the presence of inhibitor is not visible. How did authors confirm the adsorption of inhibitor on steel surface?

Reply: As can be noted from Figure 10 in the revised version the peak associated with carbon is much intense in the presence of the inhibitor, see Figures 10f and 10h that on the bare steel sample, see Figure 10b therefore, this may be indicated of the presence of the inhibitors on the steel sample surface. In the revised version, we have added the following to clarify this point.

...are not observed in the chemical analysis (Figures 10f and 10h). However, as can be noted from Figure 10 in the revised version the peak associated with carbon is much intense in the presence of the inhibitor, see Figures 10f and 10h that on the bare steel sample, see Figure 10b therefore, this may be indicated of the presence of the inhibitors on the steel sample surface.

3. There is no explanation of β_a and β_c values in Tafel polarization study. The sign of β_c should be negative.

Reply: We agree with this observation, thus in the revised version we have modified Table 2 as follows:

Table 2. Electrochemical parameters obtained by means of polarization curves for 1,2,3-triazoles 1,4-disubstituted in API 5L X52 steel immerse in HCl 1M

Inhibitor	C (ppm)	E_{corr} (mV) vs Ag/AgCl	β_c (mV dec ⁻¹)	β_a (mV dec ⁻¹)	i_{corr} (mA/cm ²)	η_{pol} (%)
Blank	0	-421.2	-106.5	84.6	0.32	-
3	5	-450.8	-107.0	54.3	0.04	87.3
3	50	-470.1	-109.5	75.0	0.04	86.9
4	5	-448.0	-163.4	80.4	0.08	76.6
4	50	-439.9	-158.8	74.7	0.04	87.0
5	5	-447.6	-146.8	52.6	0.07	79.4
5	50	-441.0	-110.2	66.3	0.02	71.0
6	5	-413.4	-122.1	30.5	0.02	94.3
6	50	-427.2	-136.2	48.4	0.02	93.5
7	5	-410.8	-161.2	52.1	0.03	90.3
7	50	-425.3	-132.2	69.5	0.04	88.2

4. The range for polarization ± 500 mV is very high. Is there any reason for this? The sweep rate should be 0.1mV/sec.

Reply: About the sweep rate, in literature it is recommended to use 0.1 to 1 mV s⁻¹, as a good practice.

5. In EIS study, Bode plot should be provided to understand the adsorption behavior of inhibitors.

Reply: We agree with this observation. Therefore, in the revised version we have included the bode plots.

...The Bode plots for API 5L X52 steel in the absence and presence of various concentrations of 1,2,3-triazoles 1,4-disubstited are represented in Fig. 5. An ideal capacitor is characterized by a fixed value of slope (unity) and phase angle (-90°). The increase in the values of phase angle in presence of different inhibitor concentrations suggest that surface roughness of API 5L X52 steel decreased due to formation of protective film by theophylline-triazole derivative [44]. The result indicated that there was two-time constant coupled, and the system could be described by three resistances which consists of electrolyte resistance (R_s), charge transfer resistance (R_{ct}), organic molecules adsorbed resistance (R_{mol}) and double layer capacitance, as shown in Fig 2b. While, in absence inhibitor the phase angle vs log frequency shows one-time constant attributed to charge transfer resistance [45]. Finally, the corrosion inhibition effectiveness of 1,2,3-triazoles 1,4-disubstited can also be interpreted in the real impedance values axis of the Bode modulus plots which also increases as the concentration increase (Fig. 5 a, c, e, g and h)...

[44] Ebenso EE, Kabanda MM. 2012. Electrochemical and quantum chemical investigation of some azine and thiazine dyes as potential corrosion inhibitors for mild steel in hydrochloric acid solution. *Ind. Eng. Chem. Res.* 51, 12940–12958. *Ind. Eng. Chem. Res.* 51, 12940–12958.

[45] Zhang H, Pang X, Zhou M, Liu C, Wei L, Gao K. 2015. The behavior of pre-corrosion effect on the performance of imidazoline-based inhibitor in 3 wt.% NaCl solution saturated with CO₂, *Appl. Surf. Sci.* 356: 63–72.

Figure 5 Bode diagrams of theophylline-triazoles compounds: a) 3, b) 4, c) 5, d) 6 and e) 7 immersed API 5L X52 steel at different inhibitor concentrations.

6. What is r_{mol} ? Is it film resistance? Clarify

Reply: The R_{mol} term refers to the “resistance of the organic molecules adsorbed” usually employed for different authors in papers related with inhibitors. Even when some authors name this value as “film resistance”, we prefer not to use this term since there could be confusion with the formation of an organic film coming from the molecules, instead of an interface formed by adsorbed molecules to the metallic surface.

7. The mechanism of inhibition should be clearly discussed. Which form of the inhibitor is predominantly adsorbed: Neutral or protonated? It should be clearly stated based on DFT calculation.

Reply: We agree with this comment thus, in the revised version of our manuscript we have included a new section regarding the mechanism of inhibition

Adsorption mechanism

The corrosion inhibition of API 5L X52 in HCl 1M provoked by 1,2,3-triazoles 1,4-disubstituted molecules can be explained as follows (Figure 9): The protonated species of the organic compounds interact, throughout Columbic forces, with previously adsorbed chlorides ions (Cl^-) present on the API 5L X52 steel surface which results in physisorption of the inhibitor molecules. This way, the inhibitor molecules compete with H^+ for electrons on steel surfaces [67]. Moreover, the donation of lone electrons pairs of nitrogen atoms to the empty orbital of Fe atoms would induce a chemisorption process of the organic molecule [68] and the accumulation of negative charges on the steel surface can be transferred from the d orbital of Fe to unoccupied π^* (anti-bonding) of 1,2,3-triazoles 1,4-disubstituted molecules (retro-donation) [69], see Figure 9a and 9b. The combination of these two types of adsorption mechanisms is carried out for the compounds evaluated. However, the compound 7 (Figure 9b) showed a stronger interaction (chemisorption) by nitrogen heteroatoms that have free electron pairs with steel, demonstrated with thermodynamic analysis using the Langmuir model.

67. Hamania H, Douadia T, Daouda D, Al-Noaimic M, Rikkouha R, Chafaa S. 2017. 1-(4-Nitrophenyl-imino)-1-(phenylhydrazono)-propan-2-one as corrosion inhibitor for mild steel in 1 M HCl solution: Weight loss, electrochemical, thermodynamic and quantum chemical studies, *J. Electroanal. Chem.* 801,425–438.
68. Singh P, Quraishi MA. 2016. Corrosion inhibition of mild steel using Novel Bis Schiff's Bases as corrosion inhibitors: *Electrochemical and Surface. Measurement* 86, 114–124.
69. Haldhar R, Prasad D, Saxena A. 2018. Myristica fragrans extract as an eco-friendly corrosion inhibitor for mild steel in 0.5 M H₂SO₄ solution, *J. Environm. Chem. Eng.* 6, 2290–2301

Compound	R
3	-H
4	-F
5	-Cl
6	-Br
7	-I

Figure 9. Schematic representation of the adsorption behavior of inhibitor on API 5L X52 steel surface immersed in 1M HCl.

Quantum chemical calculations have also been included within a new section named “Theoretical assessment”. Alejandra M. Navarrete-López was responsible for the calculations.

4.1 Theoretical assessment

The presence of nitrogen as heteroatom in the inhibitor molecules provides high tendency toward protonation in aqueous acidic medium. Under these conditions the calculations were carry out for the complete set of electrons and the geometry of involved structures was fully optimized. In other words, the nitrogen atoms hold positive charge by

adding of hydrogen atoms and the solvent (water) effect is including with the Solvation Model based on Density (SMD) [33]. The results of the geometry optimization of the selected compounds in acidic medium and solvent effect are presented in Figure 11.

A very important fact in the adsorption over the metallic surface is the planar configuration of the inhibitor molecules. The frameworks of these geometries show no planar configurations for all the inhibitors. Some angles are showed in Figure 11, the halogen substitute has an important effect over the dihedral angle, particularly with Iodine.

Quantum-chemical indexes EHOMO, ELUMO, GAP, hardness η , and dipole moment are summarized in Table 4. It is known that EHOMO is often related to electron donation ability of the inhibitor molecules toward the metallic surface atoms, and it is expected that a higher EHOMO value would favor a greater charge transfer [70]. According to these results, the values of EHOMO show the following behavior: compound 7 > compound 6 > compound 5 > compound 3 > compound 4. In this case, the notorious largest EHOMO corresponds to compound 7 in line with the aforementioned experiments at low concentrations and the lowest EHOMO corresponds to compound 4 that in terms of activity is the worst inhibitor.

Some other authors (for instance [71]) have found that a smaller value of the hardness is related to greater stability of the surface-inhibitor complex formed. The trend that we got was 4 > 3 > 5 > 6 > 7, as it can be observed in Table 4. Then the complex metallic surface-compound 7 presents the best stability and inhibition efficiency at low concentrations.

Moreover, smaller values of ELUMO present better capacity of the inhibitor to accept electrons, for this property the trend with solvent was compound 7 < 4 < 5, 6 < 3. Negative values of ELUMO indicate the capability to accept electrons too. Nevertheless, there is not a clear relationship between the above last two properties.

For the overall electrophilicity and according to the computed values (see Table 5), compound 7 exhibited a better nucleophilic character with solvent effect. Whereas compound 4 was lower than the rest. Therefore, a better interaction of compound 7 with the metallic surface would be expected than for the rest of compounds, the differences are not significant.

The Hirshfeld atomic charges for the most important centers with solvent effect are summarized in Table 6. Redistribution of the charge is observed by the presence of the substituent, the atoms that have a larger negative charge, suggests that those are active centers with excess charges that it could act as a nucleophilic group. The most favorable sites for the interaction with the metal surface are the Oxygen atoms, 24 and 25, see Figure 11 and Table 6, for all the selected compounds. As it has been expected, fluorine in compound 4 is the best electro-attractor followed by Chlorine, the halogen substitutes are active sites for the inhibition. The Carbon 8, in all the compounds, is another nucleophilic center. The Carbon 13 in compound 7 also gains charge.

Experimental data reveals that compound 7 is the best inhibitor of the corrosion, the geometric structure and the parameters here studied such as EHOMO, hardness, electrophilicity and atomic charges support this fact.

Table 6. Atomic Hirshfeld charges for the different inhibitor compounds. See the Figure 11 for the atomic labels.

Number	Atom	Compound 3	Compound 4	Compound 5	Compound 6	Compound 7
1	N	-0.011	-0.005	-0.007	-0.011	-0.003
2	N	0.305	0.306	0.306	0.310	0.313
3	N	0.087	0.088	0.087	0.087	0.090
4	C	0.164	0.178	0.177	0.151	0.188
5	C	0.074	0.075	0.074	0.075	0.077
6	C	0.275	0.275	0.276	0.274	0.276
7	C	0.210	0.218	0.219	0.216	0.227

8	C	-0.020	-0.019	-0.015	-0.012	-0.013
9	C	0.020	0.040	0.041	0.044	0.036
10	C	0.027	0.048	0.048	0.050	0.001
11	C	0.015	0.026	0.024	0.013	0.029
12	C	0.015	0.026	0.023	0.012	0.027
13	C	0.017	0.094	0.036	-0.004	-0.031
14	H	0.000	-0.167	-0.110	-0.038	-0.038
15	N	0.025	0.024	0.025	0.025	0.025
16	C	0.358	0.358	0.359	0.356	0.358
17	N	0.245	0.245	0.245	0.245	0.245
18	C	0.165	0.165	0.165	0.165	0.166
19	C	0.015	0.015	0.014	0.015	0.015
20	N	0.165	0.165	0.165	0.165	0.165
21	C	0.213	0.213	0.212	0.215	0.213
22	N	0.127	0.127	0.127	0.127	0.127
23	C	0.271	0.271	0.271	0.270	0.271
24	O	-0.353	-0.354	-0.352	-0.343	-0.352
25	O	-0.411	-0.411	-0.410	-0.411	-0.411

(a) Front.

(a) $\theta_1 = 110.9^\circ$; $\theta_2 = 111.7^\circ$

Dihedral = 100.6°

(b) Front, F= blue
Dihedral = 97.3°

(b) $\theta_1 = 111.1^\circ$; $\theta_2 = 111.0^\circ$

(c) Front, Cl= Green
Dihedral = 96.4°

(c) $\theta_1 = 111.1^\circ$; $\theta_2 = 111.2^\circ$

(d) Front, Br= orange

(d) $\theta_1 = 110.7^\circ$; $\theta_2 = 111.6^\circ$

Figure 11. Optimized molecular structures: (a) compound 3, (b) compound 4, (c) compound 5, (d) compound 6, (e) compound 7. Carbon atoms are represented in gray ball, Nitrogen in dark blue, Oxygen in red and Hydrogen in white color. Dihedral angle among 3,7,8 and 10 atoms.

33. Frisch MJ, Trucks GW, Schlegel HB, Scuseria GE, Robb MA, Cheeseman JR, Scalmani G, Barone V, Mennucci B, Petersson GA, Nakatsuji H, Caricato M, Li X, Hratchian HP, Izmaylov AF, Bloino J, Zheng G, Sonnenberg JL, Hada M, Ehara M, Toyota K, Fukuda R, Hasegawa J, Ishida M, Nakajima T, Honda Y, Kitao O, Nakai H, Vreven T, Montgomery Jr JA, Peralta JE, Ogliaro F, Bearpark M, Heyd JJ, Brothers E, Kudin KN, Staroverov VN, Kobayashi R, Normand J, Raghavachari K, Rendell A, Burant JC, Iyengar SS, Tomasi J, Cossi M, Rega N, Millam JM, Klene M, Knox JE, Cross JB, Bakken V, Adamo C, Jaramillo J, Gomperts R, Stratmann RE, Yazyev O, Austin AJ, Cammi R, Pomelli C, Ochterski JW, Martin RL, Morokuma K, Zakrzewski VG, Voth GA, Salvador P, Dannenberg JJ, Dapprich S, Daniels AD, Farkas Ö, Foresman JB, Ortiz JV, Cioslowski J, Fox DJ. 2009. Gaussian 09 (Gaussian, Inc., Wallingford CT).
70. Khaled KF. 2008. Molecular simulation, quantum chemical calculations and electrochemical studies for inhibition of mild Steel by triazoles. *Electrochim. Acta*, 53, 3484-3492.
71. Sahin M, Gece G, Kaerci F, Bligic S. Experimental and theoretical study of the effect of some heterocyclic compounds on the corrosion of low carbon steel in 3.5% NaCl medium. *J. Appl. Electrochem.*, 38, 809-815.

8. *Adsorption isotherm should be redrawn by selecting proper values on X and Y axis*

Reply: We agree with this suggestion thus, we have recalculated and converted the value on X and Y to millimolar as follows:

9. Fig 8: EDX is reported for 5 and 20 ppm concentration while optimum concentration is 50 ppm. Is there any reason for using lower concentration?

Reply: In this case, we wanted to compare compounds 5 and 7 by using the same concentration (20 ppm) because they had practically the same efficiency value. Actually, we did not compare with 5 ppm.

10. What is the significance of using API5LX52 and 1M HCl?

Reply: API 5L X52 steel was used because it is a metal that is used in the oil industry, while HCl was studied due to its aggressive behavior, even more severe than the usual conditions in plant.

Reviewer: 3

Comments to the Author(s)

COMMENTS ON MANUSCRIPT RSOS-181738

TITLE: Adsorption behavior of new theophylline-triazole based derivatives as effective corrosion inhibitors for steel in acidic medium

COMMENTS

The manuscript reports on the synthesis, characterization of theophylline-triazole based derivatives and evaluation of corrosion inhibition performance for API 5L X52 steel corrosion in 1 M HCl solution. The corrosion inhibition effect was investigated using electrochemical impedance spectroscopy and potentiodynamic polarization techniques complemented by surface morphological characterization of the corroded carbon steel samples without and with the inhibitors with SEM/EDS. The manuscript is of interest in the fields of materials and corrosion. The synthesis and characterization of the compounds used as inhibitors is well treated but the same cannot be said of the corrosion inhibition behavior of the synthesized compounds. In its present form, the manuscript is not recommended for publication in Royal Society Open Science. Major revision of the paper is needed before consideration for publication. The issues to be addressed by the authors are appended below:

(1)The Title of the manuscript should be modified to read "Adsorption and corrosion inhibition behavior of new theophylline-triazole based derivatives for steel in acidic medium"

Reply: We agree with this suggestion therefore we have modified the title as follows:

Adsorption and corrosion inhibition behavior of new theophylline-triazole based derivatives for steel in acidic medium

(2) There are important keywords not listed and the irrelevant ones are listed. Words like Acid, Theophylline, Corrosion inhibition, Steel should form part of the keywords if required by the journal.

Reply: We agree with this observation. In final version, we changed the Keywords as follows:

Keywords: Teophylline, corrosion inhibition, API 5L X52 steel, acid

(3) On page 2 line 14, 'adsorpted' should be corrected to 'adsorbed'. Also in line 24, 'absord' should be corrected to 'adsorb'

Reply: These mistakes have been corrected in the revised version.

... It has been found that molecules with lone electron pairs and/or π -electrons in their structure shows great affinity to be adsorbed into the metallic material [9-12]...

...Several reports have highlighted the capabilities of these compounds to strongly adsorb on metal surfaces, achieving an adequate corrosion inhibition efficiency at low concentrations [25-32]...

(4) The idea that the synthesized compounds were tested as 'environmentally friendly' is rather speculative and not based on experimental evidence as they were not tested experimentally to ensure that they meet the three criteria of a compound to be designated as 'environmentally friendly' which are toxicity, biodegradability and bioaccumulation. Hence the statement in line 31 of page 2 should be revised to clear this ambiguity.

Reply: We agree with this observation; therefore, we have eliminated the ambiguity and speculation by removing the 'environmentally friendly' label from the manuscript.

...These novel compounds were tested as corrosion inhibition species in order to establish structure-activity relationships and to gain further insight into their adsorption properties and the steel protection...

(5) It is not clear if the authors had check the purity of the synthesized compounds given that the melting point range is large as could be seen as follows: Compound 2 (220-226 °C); Compound 3 (169-172 °C), Compound 4 (182-186 °C); Compound 6 (194-198 °C.) Compound 7 (180-184 °C.).

Reply: We have revised and corrected the melting points of all the compounds.

1.1.1. ...1,3-dimethyl-7-(prop-2-in-1-yl)-3,7-dihydro-1H-purine-2,6-dione (2).

Compound 2 was synthesized following the procedure described by Ruddaraju et al [21]. A mixture of theophylline (1) (1.98 g, 11 mmol) and potassium carbonate (1.990 g, 14.4 mmol) in DMF (30 mL) were stirred vigorously at room temperature for 20 minutes. After this time, propargyl bromide (1.68 mL, 22.2 mmol) was added and temperature

was increased at 85 °C with vigorous stirring for another 2 h. Then, the mixture was poured in cold water. The compound was recovered as a white powder: yield 80%, m.p. 220-222 °C. ¹H NMR (500.13 MHz, CDCl₃): δ (ppm)= 2.60 (1H, t, J=2.61 Hz, H12), 3.41 (3H, s, N1-CH₃), 3.60 (3H, s, N3-CH₃), 5.17 (2H, dd, J= 2.6, 0.62 Hz, H10), 7.83 (1H, t, J=0.53 Hz, H8). ¹³C NMR (125.77 MHz, CDCl₃): δ= 27.96 (N1-CH₃), 29.79 (N3-CH₃), 36.44 (C10), 75.43 (C11), 76.07 (C12), 106.71 (C5), 140.42 (C8), 148.92 (C4), 151.60 (C2), 155.23 (C6). FT-IR/ATR v_{max}/cm⁻¹: 3243.55, 3111.71, 2946.11, 2127.13, 1703.88, 1651.15, 1543.95, 1477.34, 1437.21, 1373.59, 1232.32, 1190.89, 1025.01, 977.45, 744.20.

3.1.2. 7-((1-benzyl-1H-1,2,3-triazol-4-yl) methyl)-1,3-dimethyl-3,7-dihydro-1H-purine-2,6-dione (3).

A mixture of compound 2 (206 mg, 1mmol), sodium ascorbate (40 mg, 0.2 mmol), sodium azide (78 mg, 1.2 mmol), benzyl chloride (0.14 mL, 1.2 mmol), and Cu/Al mixed oxide (40 mg) in 6 mL of ethanol/water (3:1) were stirred at 80 °C for 30 min with microwave radiation. After this time, the Cu/Al mixed oxide is recovered by centrifugation and the supernatant is poured in 20 mL of water, extracted with dichloromethane, and dried over sodium sulfate anhydrous. Compound 3 is obtained, after chromatographic purification (CH₂Cl₂:EtOH 95:5), as a white powder: yield 78%, m.p. 169-171 °C. ¹H NMR (500.13 MHz, CDCl₃): δ= 3.38 (3H, s, N1-CH₃), 3.56 (3H, s, N3-CH₃), 5.49 (2H, s, H13), 5.56 (2H, s, H10), 7.26 (2H, m, H15), 7.36 (3H, m, H17, H16), 7.75 (1H, s, H12), 7.81 (1H, s, H8). ¹³C NMR (125.77 MHz, CDCl₃): δ= 27.98 (N1-CH₃), 29.81 (N3-CH₃), 41.48 (C10), 54.32 (C13), 106.45 (C5), 123.48 (C12), 128.09 (C15), 128.89 (C17), 129.15 (C16), 134.23 (C14), 141.32 (C8), 142.52 (C11), 148.93 (C4), 151.58 (C2), 155.40 (C6). FT-IR/ATR v_{max}/cm⁻¹: 3114.70, 2957.28, 1690.39, 1650.26, 1546.81, 1453.58, 1214.63, 1021.75, 749.91. HRMS (ESI-TOF) (calculated for C₁₇H₁₈N₇O₂ + H⁺): 352.1516; found: 352.1514.

3.1.3. 7-((1-(4-fluorobenzyl)-1H-1,2,3-triazol-4-yl) methyl)-1,3-dimethyl-3,7-dihydro-1H-purine-2,6-dione (4).

Compound 4 was synthesized following the procedure described previously for compound 3, from compound 2 and 4-fluorobenzyl chloride. Compound 4 is obtained, after chromatographic purification (CH₂Cl₂:EtOH 95:5), as a white powder: yield 90%, m.p. 184-186 °C. ¹H NMR (400.13 MHz, CDCl₃): δ= 3.39 (3H, s, N1-CH₃), 3.56 (3H, s, N3-CH₃), 5.47 (2H, s, H13), 5.56 (2H, s, H10), 7.06 (2H, t, J= 8.61 Hz, H15), 7.26 (2H, dd, J= 8.64, 4.34 Hz, H16), 7.75 (1H, s, H12), 7.82 (1H, s, H8). ¹³C NMR (100.61 MHz, CDCl₃): δ= 27.99 (N1-CH₃), 29.82 (N3-CH₃), 41.47 (C10), 53.58 (C13), 106.45 (C5), 116.09 (C15), 116.31 (C15), 123.39 (C12), 129.98 (C16), 130.06 (C16), 141.35 (C8), 142.65 (C11), 148.99 (C4), 151.59 (C2), 155.44 (C6), 161.69 (C14 or C17), 164.17 (C14 or C17). FT-IR/ATR v_{max}/cm⁻¹: 3144.88, 3116.27, 3000.48, 2960.31, 1691.08, 1651.91, 1549.09, 1512.06, 1456.51, 1226.98, 1023.54, 786.67, 750.59, 615.31, 522.38. HRMS (ESI-TOF) (calculated for C₁₇H₁₇N₇O₂F + H⁺): 370.1422; found: 370.1419.

3.1.4. 7-((1-(4-chlorobenzyl)-1H-1,2,3-triazol-4-yl) methyl)-1,3-dimethyl-3,7-dihydro-1H-purine-2,6-dione (5).

Compound 5 was synthesized following the procedure described for compound 3, from compound 2 and 4-chlorobenzyl chloride. Compound 5 is obtained, after chromatographic purification (CH₂Cl₂: EtOH 95:5), as a light green powder: yield 76%, m.p. 194-196 °C. ¹H RMN (400.13 MHz, CDCl₃): δ= 3.39 (3H, s, N1-CH₃), 3.56 (3H, s, N3-CH₃), 5.47 (2H, s, H13), 5.56 (2H, s, H10), 7.20 (2H, d, J= 8.42 Hz, H15), 7.34 (2H, d, J= 842. Hz, H16), 7.77 (1H, s, H12), 7.82 (1H, s, H8). ¹³C NMR (100.61 MHz, CDCl₃): δ= 27.98 (N1-CH₃), 29.81 (N3-CH₃), 41.45 (C10), 53.58 (C13), 106.45 (C5), 123.53 (C12), 129.38 (C15), 129.44 (C16), 132.72 (C14), 135.01 (C17), 141.36 (C8), 142.77 (C11), 149 (C4), 151.58 (C2), 155.44 (C6). FT-IR/ATR v_{max}/cm⁻¹: 3096.88, 3052.15, 2960.28, 1688.25, 1650.83, 1555.24, 1406.79, 1220.94, 1082.17, 1045.89, 978.31, 848.97, 785.63, 770.92, 608.01, 494.81. HRMS (ESI-TOF) (calculated for C₁₇H₁₇N₇O₂Cl + H⁺): 386.1127; found: 386.1124.

3.1.5. 7-((1-(4-bromobenzyl)-1H-1,2,3-triazol-4-yl) methyl)-1,3-dimethyl-3,7-dihydro-1H-purine-2,6-dione (6).

Compound 6 was synthesized following the procedure described for compound 3, from compound 2 and 4-bromobenzyl bromide. Compound 6 is obtained, after chromatographic purification (CH₂Cl₂:EtOH 95:5), as a white powder: yield 63%, m.p. 199-201 °C. ¹H RMN (500.13 MHz, CDCl₃): δ= 3.39 (3H, s, N1-CH₃), 3.56 (3H, s, N3-CH₃), 5.45 (2H, s, H13), 5.56 (2H, s, H10), 7.13 (2H, d, J= 8.65 Hz, H15), 7.49 (2H, d, J= 8.61 Hz, H16), 7.76 (1H, s, H12), 7.80 (1H, s, H8). ¹³C NMR (125.77 MHz, CDCl₃): δ= 27.96 (N1-CH₃), 29.79 (N3-CH₃), 41.46 (C10), 53.62 (C13), 106.45 (C5), 123.09 (C17), 123.49 (C12), 129.69 (C15), 132.33 (C16), 133.24 (C14), 141.31 (C8), 142.81 (C11), 149 (C4), 151.56 (C2), 155.42 (C6). FT-IR/ATR v_{max}/cm⁻¹: 3134.60, 3095.80, 3049.46, 2960.09, 1687.19, 1650.07, 1554.67, 1450.68, 1405.55, 1220.14, 1103.66, 1030.53, 978.62, 847.88, 607.67, 488.92. HRMS (ESI-TOF) (calculated for C₁₇H₁₇N₇O₂Br + H⁺): 430.0621; found: 430.0618.

3.1.6. 7-((1-(4-iodobenzyl)-1H-1,2,3-triazol-4-yl) methyl)-1,3-dimethyl-3,7-dihydro-1H-purine-2,6-dione (7).

Compound 7 was synthesized following the procedure described for compound 3, from compound 2 and 4-iodobenzyl bromide. Compound 7 is obtained, after chromatographic purification (CH₂Cl₂:EtOH 95:5), as a white powder: yield 68%, m.p. 181-184 °C. ¹H NMR (400.13 MHz, CDCl₃): δ= 3.39 (3H, s, N1-CH₃), 3.56 (3H, s, N3-CH₃), 5.44 (2H, s, H13), 5.56 (2H, s, H10), 7.00 (2H, d, J= 8.38 Hz, H15), 7.69 (2H, d, J= 8.37 Hz, H16), 7.76 (1H, s, H12), 7.82 (1H, s, H8). ¹³C NMR (100.61 MHz, CDCl₃): δ= 28.02 (N1-CH₃), 29.84 (N3-CH₃), 41.46 (C10), 53.73 (C13), 94.76 (C17), 106.44 (C5), 123.54 (C12), 129.88 (C15), 133.88 (C14), 138.29 (C16), 141.35 (C8), 142.72 (C11), 148.98 (C4), 151.58 (C2), 155.43 (C6). FT-IR/ATR v_{max}/cm⁻¹: 3143.55, 3115.45, 2958.37, 1690.45, 1651.19, 1548.13, 1456.04, 1218.88, 1006.66, 750.30, 615.40, 498.95. HRMS (ESI-TOF) (calculated for C₁₇H₁₇N₇O₂I + H⁺): 478.0483; found: 478.0478.

(6) The discussion on EIS measurements results should be rewritten. It is very shallow and should be expanded to convey meaning. Fundamental questions like the need of using constant phase element in the equivalent circuit need to be answered. Also for the purpose of clarity, the authors need to use appropriate scientific words to describe the technique. It is not clear what the authors meant by "the results were adjusted using equivalent electric circuits"

Reply: In the final version, we have modified the discussion in the results section.

4.1 OCP and EIS Electrochemical evaluation

Fig. 3a shows the variations in time of the open circuit potential (OCP) in static conditions at 50 ppm, at the electrode made of API 5L X52 steel in presence of different inhibitor concentrations. It should be mentioned that it was necessary the stabilization of the OCP previously to do the EIS determinations. The steady state was reached after 1700 seconds. Fig. 3b corresponds to the Nyquist diagram for the system without inhibitor, which depicts a depressed semicircle reaching a maximum Z_{real} value of 30 Ω cm² (adjusted with electrical circuit Figure 2a). The Nyquist diagrams of each of the theophylline-triazoles are shown in Figs. 4a-4e. As can be seen, the diameter of the semicircle increases proportionally with the inhibitor's concentration. According to the shape of the semicircle, two time constant (using the circuit figure 2b) can be attributed, the charge transference resistance and the second to adsorbed molecules resistance [36-38]. The depressed semicircle is attributed due to frequency dispersion effect and surface irregularities and heterogeneities [39].

It can be observed that the Z_{real} value presents a large variation. Based on these results, it can be inferred that the presence of a halogen substituent in para position of the aromatic ring can modulate the inhibition capacity of the compound.

After obtaining the Nyquist diagrams of the compounds at different concentrations, the impedance results for API 5L X52 steel with and without inhibitor can be explained by an equivalent circuit (Figure 2) which comprise of R_{ct}

(charge transfer resistance), and Q is the constant phase element in parallel with R_s (solution resistance), R_{mol} is the molecules resistance.

Constant phase elements have widely been used [40] to account for deviations brought about by surface roughness. The impedance of CPE is given by the next equation [41]:

$$Z_{CPE} = Q^{-1}(j\omega)^{-n} \quad (1)$$

Where Y_0 is the magnitude of the CPE, n the CPE exponent (phase shift), ω the angular frequency ($\omega=2\pi f$, where f is the AC frequency), and j here is the imaginary unit. The capacity correction to its real values was calculated from Eq. (2), where ω_{max} is the frequency at which the imaginary part of the impedance ($-Z_{imag}$) has a maximum. C_{dl} represents the double layer capacitance (Eq. 2).

$$C_{dl} = Y_0(\omega_m^n)^{n-1} \quad (2)$$

The value of inhibition efficiency (η) can be obtained by means of the following equation [42-43]:

$$\eta (\%) = \frac{\left(\frac{1}{R_p}\right)_{blank} - \left(\frac{1}{R_p}\right)_{inhibitor}}{\left(\frac{1}{R_p}\right)_{blank}} \times 100 \quad (3)$$

36. Obayes HR, Al-Amiery AA, Alwan G H, Amir A, Kadhum H, Mohamad A B. 2017. Sulphonamides as corrosion inhibitor: Experimental and DFT studies. *J. of Mol. Struct.* 1138, 27-34
37. Obot IB, Ankah NK, Sorour AA, Gasem ZM, Haruna K. 2017. 8-Hydroxyquinoline as an alternative green and sustainable acidizing oilfield corrosion inhibitor. *Sust. Mat. and Techn.*, 14, 1-10
38. Pfeiffer M, Klock H, Helmut, Bergen G. Ehrenhaft, Ferreira P, Gollnick J., Fischer CB. 2017. Surface protection of low carbon steel with N-acyl sarcosine derivatives as green corrosion inhibitors. *Surf. Interfaces*, 9, 70-78.
39. Qiang Y, Zhang S, Yan S, Zou X, Chen S. 2017. Three indazole derivatives as corrosion inhibitors of copper in a neutral chloride solution. *Corros. Sci.* 126, 295-304.
40. Yadav M, Behera D, Sharma U. 2016. Nontoxic corrosion inhibitors for N80 steel in hydrochloric acid. *Arab. J. of Chem.*, 9(2), s1487-s1495.
41. Mobin M, Basik M, Aslam J. 2018. Boswellia serrata gum as highly efficient and sustainable corrosion inhibitor for low carbon steel in 1 M HCl solution: Experimental and DFT studies. *J. Mol. Liq.*, 263, 174–186
42. Javadian S, Yousefi A, Neshati J (2013) Synergistic effect of mixed cationic and anionic surfactants on the corrosion inhibitor behavior of mild steel in 3.5% NaCl. *Appl. Surf. Sci.*, 285, 674–681.
43. Zarrouk A, Zarrouk H, Ramli Y. (2016) Inhibitive properties, adsorption and theoretical study of 3,7-dimethyl-1-(prop-2-yn-1-yl)quinoxalin-2(1H)-one as efficient corrosion inhibitor for carbon steel in hydrochloric acid solution. *J. Mol. Liq.* 222, 239–252.

(7) The authors should also present the EIS results in Bode formats as the number of time constants is better revealed in Bode representation than in Nyquist representation.

Reply: Reply: We agree with this observation. Therefore, in the revised version we have included the bode plots.

...The Bode plots for API 5L X52 steel in the absence and presence of various concentrations of 1,2,3-triazoles 1,4-disubstited are represented in Fig. 5. An ideal capacitor is characterized by a fixed value of slope (unity) and phase angle (-90°). The increase in the values of phase angle in presence of different inhibitor concentrations suggest that

surface roughness of API 5L X52 steel decreased due to formation of protective film by theophylline-triazole derivative [43]. The result indicated that there was two-time constant coupled, and the system could be described by three resistances which consists of electrolyte resistance (R_s), charge transfer resistance (R_{ct}), organic molecules adsorbed resistance (R_{mol}) and double layer capacitance, as shown in Fig 2b. While, in absence inhibitor the phase angle vs log frequency show one time constant attributed to charge transfer resistance [44]. Finally, the corrosion inhibition effectiveness of 1,2,3-triazoles 1,4-disubstited can also be interpreted in the real impedance values axis of the Bode modulus plots which also increases as the concentration increase (Fig. 5 a,c, e, g and h)...

[43] Ebenso EE, Kabanda MM. 2012. Electrochemical and quantum chemical investigation of some azine and thiazine dyes as potential corrosion inhibitors for mild steel in hydrochloric acid solution. *Ind. Eng. Chem. Res.* 51, 12940–12958.

[44] Zhang H, Pang X, Zhou M, Liu C, Wei L, Gao K. 2015. The behavior of pre-corrosion effect on the performance of imidazoline-based inhibitor in 3 wt.% NaCl solution saturated with CO₂, *Appl. Surf. Sci.* 356: 63–72.

(8) Again, the discussion on potentiodynamic polarization technique should be rewritten as it is very shallow. The authors have not discussed the Fig.6 at all. The scope of the discussion should be expanded to make meaning.

Reply: We agree with this observation. The discussion on potentiodynamic polarization technique was added in the revised version as follows:

... Fig. 6 shows that all of the curves shift to a lower current density for both the anodic and cathodic half-reactions with the addition of 1,2,3-triazoles 1,4-disubstituted in API 5L X52 steel immersed in HCl 1M, and the trend is more pronounced at 5 ppm inhibitor concentration; indicating that the anodic dissolution of API 5L X52 steel and cathodic reduction of hydrogen ions were inhibited [50].

This suggests that the rate of electrochemical reaction was reduced due to the formation of a protective layer of inhibitor molecules over the steel surface [51].

Furthermore, the cathodic Tafel curves are parallel (Fig. 5a and 5b), which shows that there is no change in the hydrogen evolution mechanism with the addition of 1,2,3-triazoles 1,4-disubstituted. and the reduction of hydrogen ions mainly takes place through a charge transfer [52]...

[50] Quraishia MA, Lgazd H, Salghi R. 2018, Thiosemicarbazide and thiocarbohydrazide functionalized chitosan asecofriendly corrosion inhibitors for carbon steel in hydrochloric acid solution, *Int. J. Biol. Macrom.*, 107,1747–1757.

[51] Stoyanova A, Petkova G, Peyerimhoff S, 2002, Correlation between the molecularstructure and the corrosion inhibiting effect of some pyrophthalonecompounds, *Chem. Phys.* 279 (1), 1–6.

[52] Solmaz R, Kardas G, Culha M, Yazıcı B, Erbil M, 2008, Investigation of adsorptionand inhibitive effect of 2-mercaptothiazoline on corrosion of mild steel inhydrochloric acid media, *Electrochim. Acta* 53 (20), 5941–5952.

(9) In Table 1, the authors should replace the SD with Chi squared values (values of goodness of fit) to ascertain the validity of the equivalent circuit used to fit the experimental data.

Reply: In final version, we changed the SD values with chi squared values

Table 1. Electrochemical parameters of 1,2,3-triazoles 1,4-disubstituted in API 5L X52 steel immerse in HCl 1M

Inhibitor	C (ppm)	Rs (Ω cm ²)	n	C _{dl} (μ F cm ⁻²)	R _{ct} (Ω cm ²)	R _{mol} (Ω cm ²)	η (%)	Chi squared
Blank	0	0.8	0.8	310.0	30	-	-	-
	5	1.1	0.72	221.9	154.0	5.1	81.1	0.0833
3	10	1.2	0.69	230.7	227.6	13.4	87.6	0.1119
	20	1.0	0.67	205.1	286.1	16.2	90.1	0.1130
	50	1.1	0.68	195.4	343.1	17.3	91.7	0.1052
4	5	1.1	0.73	228.9	49.9	9.6	49.6	0.1447
	10	1.3	0.75	192.8	112.4	18.9	77.2	0.1672
	20	1.2	0.72	193.4	168.1	19.5	84.0	0.1415
	50	1.2	0.71	182.7	210.4	18.6	86.9	0.1366
5	5	11.5	0.74	247.3	35.4	36.8	58.4	0.0020
	10	8.0	0.88	214.8	68.1	16.7	64.6	0.0016
	20	8.5	0.95	170.8	265.7	5.4	88.9	0.0017
	50	6.3	0.62	89.9	492.8	6.7	94.0	0.0014
6	5	1.0	0.70	174.4	134.5	14.4	79.9	0.1245
	10	1.1	0.72	129.4	169.8	29.8	85.0	0.0893
	20	1.1	0.68	123.2	248.4	28.7	89.2	0.1065
	50	1.2	0.63	108.2	335.3	29.5	91.8	0.1065
7	5	1.5	0.70	172.7	201.8	5.1	85.5	0.0024
	10	1.0	0.65	153.4	224.1	0.5	86.6	0.0036
	20	0.9	0.63	158.1	272.4	2.6	89.1	0.0045
	50	1.0	0.60	153.2	328.2	2.9	90.9	0.0040

(10) The units of bc and ba are not correct. The units should be mV/dec

Reply: We agree with this observation. In the final version, we changed the units as follows:

Table 2. Electrochemical parameters obtained by means of polarization curves for 1,2,3-triazoles 1,4-disubstituted in API 5L X52 steel immerse in HCl 1M

Inhibitor	C (ppm)	E_{corr} (mV) vs Ag/AgCl	bc ($mV dec^{-1}$)	ba ($mV dec^{-1}$)	i_{corr} (mA/cm^2)	η_{pol} (%)
Blank	0	-421.2	-106.5	84.6	0.32	-
3	5	-450.8	-107.0	54.3	0.04	87.3
3	50	-470.1	-109.5	75.0	0.04	86.9
4	5	-448.0	-163.4	80.4	0.08	76.6
4	50	-439.9	-158.8	74.7	0.04	87.0
5	5	-447.6	-146.8	52.6	0.07	79.4
5	50	-441.0	-110.2	66.3	0.02	71.0
6	5	-413.4	-122.1	30.5	0.02	94.3
6	50	-427.2	-136.2	48.4	0.02	93.5
7	5	-410.8	-161.2	52.1	0.03	90.3
7	50	-425.3	-132.2	69.5	0.04	88.2

(11) In Table 3, the authors should the replace linear regression equation with values of the slopes. Also values of K_{ads} and not $\ln K_{ads}$ should be given.

Reply: We agree with this suggestion thus Table 3 has been modified accordingly.

Table 3. Adjustment of thermodynamic data with the Langmuir isotherm

Compound	K_{ads}	$\Delta G^\circ ads$ ($kJ mol^{-1}$)	Slopes (M)	R^2
3	2.66×10^7	-41.7	$C / \theta = 1.0.759C$	0.9982
4	5.95×10^6	-38.1	$C / \theta = 1.0788C$	0.9979
5	5.38×10^6	-37.8	$C / \theta = 0.9749C$	0.9965
6	2.66×10^7	-41.7	$C / \theta = 1.0700C$	0.9998
7	5.37×10^7	-43.4	$C / \theta = 1.0895C$	0.9989

(12) Finally the English language of this manuscript is very poor and need serious improvement. The assistance of native English speakers may be sought in this regards.

Reply: The English of the manuscript was thoroughly revised.